

# Empirical model of multiple scattering effect on single-wavelength lidar data of aerosols and clouds

Valery Shcherbakov[1,2], Frédéric Szczap[1], Alaa Alkasem[1], Guillaume Mioche[1,2], Céline Cornet[3]

[1]Université Clermont Auvergne, CNRS, UMR 6016, Laboratoire de Météorologie Physique, 63178 Aubière, France
[2]Université Clermont Auvergne, Institut Universitaire de Technologie Clermont Auvergne – site de Montluçon, 03100 Montluçon, France
[3]Université Lille 1, CNRS, UMR 8518, Laboratoire d'Optique Atmosphérique, 59655 Villeneuve d'Ascq, France

*Correspondence to*: Valery Shcherbakov (v.shcherbakov@opgc.univ-bpclermont.fr)

**Abstract.** We performed extensive Monte Carlo (MC) simulations of single-wavelength lidar signals from a plane-parallel homogeneous layer of atmospheric particles and developed an empirical model to account for the multiple scattering in the lidar signals. The simulations have taken into consideration four types of lidar configurations (the ground based, the airborne, the CALIOP, and the ATLID) and four types of particles (coarse aerosol, water cloud, jet-stream cirrus and cirrus). Most of

simulations were performed with the spatial resolution of 20 m and the particles extinction coefficient $\varepsilon_p$ between 0.06 km[-1] and 1.0 km[-1]. The resolution was of 5 m for high values of $\varepsilon_p$ (up to 10.0 km[-1]). The majority of simulations for ground-based and airborne lidars were performed at two values of the receiver field-of-view (RFOV): 0.25 mrad and 1.0 mrad. The effect of the width of the RFOV was studied for the values up to 50 mrad.

The proposed empirical model is a function that has only three free parameters and approximates the multiple-scattering

relative contribution to lidar signals. It is demonstrated that the empirical model has very good quality of MC data fitting for all considered cases.

Special attention was given to the usual operational conditions, i.e., low distances to a particles layer, small optical depths and quite narrow receiver field-of-views. It is demonstrated that multiple scattering effects cannot be neglected when the distance to a particles layer is about 8 km or higher and the full RFOV is of 1.0 mrad. As for the full RFOV of 0.25 mrad, the single

scattering approximation is acceptable for aerosols ($\varepsilon_p \lesssim 1.0$ km[-1]), water clouds ($\varepsilon_p \lesssim 0.5$ km[-1]), and cirrus clouds ($\varepsilon_p \leq 0.1$ km[-1]). When the distance to a particles layer is of 1 km, the single scattering approximation is acceptable for aerosols and water clouds ($\varepsilon_p \lesssim 1.0$ km[-1], both RFOV = 0.25 and RFOV = 1 mrad). As for cirrus clouds, the effect of multiple scattering cannot be neglected even at such low distance when $\varepsilon_p \gtrsim 0.5$ km[-1].



# 1 Introduction

It is well accepted that single-wavelength lidar signals from cloud or aerosol layers are affected by multiple scattering (MS) when the optical thickness is quite high or/and the distance to a layer is large (see, e.g., Winker and Poole, 1995; Bissonnette et al., 1995; Winker, 2003). A large footprint of the receiver field-of-view (RFOV) is usually referred to as an intuitive justification of the multiple-scattering importance for signals of spaceborne lidars (see, e.g., Winker and Poole, 1995; Winker, 2003). For example, the Cloud-Aerosol Lidar with Orthogonal Polarization (CALIOP) has the footprint about 90 m (Winker

et al., 2010), which is "roughly two orders of magnitude larger than for ground-based or airborne lidars, due to the large distance from the atmosphere, allowing a much greater fraction of the multiply-scattered light to contribute to the return signal" (Winker, 2003). It follows from Monte-Carlo simulations of CALIOP signals that multiple scattering is of importance even though photon mean free paths are much larger than the footprint diameter, e.g., cirrus clouds or aerosol layers (see, e.g., Winker, 2003).

If the distance to a cloud or an aerosol layer is low, the footprint is rather small. For example, for the typical RFOV of 0.25 mrad and the distance to a layer of 8 km the footprint diameter is of 2 m; if the RFOV is of 1.0 mrad, the footprint diameter is of 8 m. (Note that in this work RFOV refers to the full angle.) If the distance to a layer is of 1 km, e.g., an airborne lidar, the footprint diameters become of 0.25 and 1 meter, respectively. Intuitively, one may expect that the effect of multiple scattering on lidar signals can be neglected with such low footprints and when the extinction coefficient of turbid medium is quite low,

for example, 1.0 km$^{-1}$ or lower. On the other hand, RFOV "can never be infinitely small to satisfy the single scattering condition" (Bissonnette, 2005). In addition, "the nature of the multiple scattering is fundamentally dependent on the scattering phase function of the atmospheric particles" (Winker, 2003). Thus, the applicability of the single scattering approximation to lidar signals from layers of large particles, e.g., cirrus clouds, can be suspect.

A number of approximate models, i.e., non-Monte-Carlo approaches to simulate lidar signals in multiple scattering conditions

were developed from 1970's to 2010's (see, e.g., Bissonnette, 2005 and references therein; Eloranta, 1998; Hogan, 2008; Hogan and Battaglia, 2008). A detailed analysis of those approaches is beyond the scope of this work. We only underscore that they are physically-based, that is, some kind of simplifications or/and approximations are employed, e.g., the time-dependent two-stream approximation (Hogan and Battaglia, 2008). Usually, the approximate models accept varying profiles or multiple layers of cloud and aerosol, and they are very fast as compared to Monte Carlo simulations. Moreover, the corresponding software,

e.g., of the models by Eloranta (1998), Hogan (2008), and Hogan and Battaglia (2008) is freely available. At the same time, we believe that the accuracy level and the applicability bounds of the approximate models still need to be rigorously evaluated. Some works devoted to Monte Carlo (MC) simulations of signals of ground-based lidars were performed from 70's to 90's (see, e.g., Plass and Kattawar, 1971; Kunkel and Weinman, 1976; Platt, 1981; Bissonnette et al. 1995; Ackermann et al., 1999). It was demonstrated that multiple scattering affects lidar signals. At the same time, it should be mentioned that those

simulations were performed in conditions that were favourable for multiple scattering: either with a high extinction coefficient





(10 km$^{-1}$ or higher) or with a large RFOV (4 mrad or larger). In the 21st century, the focus of interest of Monte Carlo simulations has mainly shifted to signals from spaceborne lidars.

As for experimental data of ground-based or airborne lidars, it is common practice to assume that multiple scattering is negligible and can be ignored. Usually, that assumption is implicitly implied or mentioned with the relation to the following

factors: a narrow RFOV, a small footprint, and a quite low value of the extinction coefficient. The only exception is cirrus clouds observed with a ground-based lidar, that is, the majority of works take into account multiple scattering employing one of possible multiple-scattering functions (MSF) (see the discussion in Appendix A) or models (see, e.g., Nakoudi et al., 2021 and references therein).

To our knowledge, there exist no works where the applicability of the single scattering approximation to lidar signals from

low distances and low optical depths was thoroughly investigated. Such an investigation is one of objectives of this work. It was performed using the Monte Carlo technics with special attention to quantitative data.

It follows from our extensive MC simulations that MS relative-contribution to lidar signals has the same general behaviour as a function of the in-cloud penetration depth when plotted as a log–linear graph. That property is valid for a wide variety of particles properties, extinction-coefficient values, and lidar configurations (see figures in Sections 5 and 6 below). Careful

analyses of figures published in the literature confirmed that conclusion. The fact that a set of simulated data have the same general behaviour suggests the idea to search for a function, which can provide a good fit to the data. Thus, the second objective of this work is to propose and test an empirical model, which can be a simple and fast tool to compute multiple-scattering effects on lidar signals.

The organization of this paper is as follows. The methodology and conditions of our Monte Carlo simulations are presented in

Section 2. Section 3 is devoted to the mathematical background and the analysis of some general features of multiple scattering impact. Our empirical model of multiple scattering effect is discussed in Section 4. Section 5 is devoted to results of our MC simulations and fittings with the empirical model for cases of low distances and small optical depths. Section 6 is devoted to cases when impact of multiple scattering is high, i.e., spaceborne lidars, high values of the extinction coefficient, and wide RFOVs. Some important methodological questions are discussed in Appendixes A and B.

## 85 2 Methodology and simulations conditions

The principal tool to simulate lidar signals was the McRALI (Monte-Carlo Radar Lidar) software developed at the Laboratoire de Météorologie Physique (Alkasem et al., 2017; Szczap et al., 2021). The software employs a forward Monte-Carlo (MC) approach along with the locate estimate method to simulate propagation of radiation (see, e.g., Marchuk et al., 2013). McRALI is based on the 3DMCPOL model (Cornet et al., 2010). The polarization state of the radiation is computed using Stokes vectors

and scattering matrixes of atmospheric compounds. It takes into account molecular scattering. In this work, the properties of the atmosphere were assigned according to the 1976 standard atmosphere (NOAA, 1976). McRALI, is a fully 3-D software, that is, values of the extinction coefficients, the single scattering albedos, and the scattering matrixes are assigned in 3D-space.





Moreover, the mixture of different types of aerosols and/or clouds is allowed. The position of a lidar can be anywhere within or outside of the atmosphere, that is, spaceborne, airborne, and ground-based measurement conditions can be simulated. A
user can assign a lidar beam direction, a RFOV, and a Stokes vector and a divergence of the emitted light. It was demonstrated in the work by Alkasem et al. (2017) that McRALI simulations are in good agreement with published results of lidar-signals modelling in multiple scattering conditions.

Four lidar configurations were taken into consideration in this work. Two configurations were monostatic coaxial zenith-looking lidars, i.e., the ground-based (the altitude is of $h = 0$ km) and the airborne ($h = 7$ km); and two values of the RFOV
were evaluated for each case, i.e., 0.25 mrad and 1.0 mrad. The emitted light was linearly polarized delta-pulse at the wavelength $\lambda$ of 0.532 μm. Its divergence was of 0.14 mrad. Essentially, we used the characteristics of the lidar system that is in operation at Clermont-Ferrand (Freville et al., 2015). For brevity sake, we will use the term the "usual operational conditions" (UOCs) when the distance from a lidar to a layer of particles is lower than 15 km, the RFOV $\leq 1$ mrad, the emitter field-of-view (EFOV) $\leq 0.2$ mrad, EFOV $\ll$ RFOV, the extinction coefficient $\varepsilon \leq 1$ km$^{-1}$. All simulations of Section 5 were
performed for the UOCs.

Other two configurations were spaceborne nadir-looking lidars. We call the "CALIOP configuration" the lidar at the altitude of 705 km having the RFOV of 0.13 mrad and the EFOV of 0.1 mrad. Only the wavelength of 0.532 μm was considered. We call the "ATLID configuration" (ATmospheric LIDar) the lidar at the altitude of 393 km having the RFOV of 0.065 mrad and the EFOV of 0.045 mrad (see, e.g., Hélière et al, 2012). Only the wavelength $\lambda = 0.355$ μm was considered. Note that the
both configurations should be considered as proxies of the real lidar systems. The objectives of this work do not require taking into account neither the pointing off-nadir nor the high-spectral-resolution separation of molecular and particulate backscattering (see, e.g., Bruneau and Pelon, 2021).

The majority of our MC data were computed so that photons were integrated over the range gate of 20 m, i.e., they correspond to photon counting mode. Such small value of the range gate was chosen with the aim to study multiple scattering in details
regardless of the fact that it does not correspond to real lidar systems. In other words, the spatial resolution of our data is of 20 m. In order to assure good statistical quality of our Monte-Carlo modelling, each signal was simulated with $4 \cdot 10^{10}$ photons emitted by the lidar (with $4 \cdot 10^{11}$ photons for the cirrus clouds having $\varepsilon = 0.06$ km$^{-1}$). Simulations of signals were performed for the orders of scattering $n = 1$ (single scattering), $n = 2$ (double scattering), and multiple scattering with $n$ equal either 20, or 40, or 50. (We have verified that the difference between data obtained with $n = 20$ and $n = 10$ was not statistically
significant for the majority of simulations conditions of this work.) In the cases of wide RFOV (Section 6.2) and high extinction coefficient (Section 6.1.2), 40 and 50, respectively, orders of scattering were considered.

The simulations of this work were performed for four types of particles, namely, a coarse-aerosol layer, a warm cloud and two types of cirrus clouds. A mixture of particles was not considered. Because Monte Carlo methods are very time consuming, our study was restricted to the case of the plane-parallel homogeneous layer placed within the altitude $h$ range of [8 – 11] km. That
range was deliberately chosen for all four types of particles despite the fact that it does not correspond to the usual altitudes of coarse aerosols or warm clouds. With such a choice, the phase-function impact on multiple scattering is free of the distance-



variation interference. It should be underlined that the results of the Sections 5 and 6 are presented so that they remain unaltered when the lidar pointing angle and/or the layer altitude vary provided that the distance to the cloud base/border remains unchanged.

The scattering matrixes were computed for the wavelengths of 0.355 μm and 0.532 μm, and the values of refractive index corresponding to the published works: (*i*) ice particles (Warren and Brandt, 2008), (*ii*) water spheres (IAPWS, 1997), and (*iii*) coarse-aerosol particles (Dubovik et al, 2002). Knowing that the multiple scattering is fundamentally dependent on the scattering matrix of particles, especially, at the small forward and backward angles, and in order to avoid effects due to quantization, all matrixes used in this work were computed with the angular resolution of 0.01 degree (about 0.175 mrad). In

addition, McRALI employs a spline interpolation to compute the cumulative distribution function. That function is used to get a random value of the scattering zenith angle for each scattering event (Cornet et al., 2010). (We have verified that MC simulations were biased when the angular resolution of a scattering matrix was worse than 0.1 degree.)

The single scattering characteristics of ice particles were computed using the Improved Geometric Optics Method (Yang and Liou, 1996); the particles are assumed to be hexagonal ice crystals having deeply-rough surface of the facets. As a consequence

of the surface roughness, the scattering matrix of ice particles has neither halo features (see, e.g., Shcherbakov, 2013) nor the delta transmission term (Yang et al., 2013). The size distribution of particles was taken to be the gamma distribution. We have considered two types of cirrus clouds that differ by the value of the effective diameter $d_{eff}$. The values $d_{eff} = 56.8$ μm (the standard deviation of 20.1 μm) and $d_{eff} = 80.0$ μm (the standard deviation of 24.5 μm) correspond to the data for jet-stream (JS) cirrus clouds and cirrus clouds (Ci), respectively, of the work by Gayet et al. (2006). The obtained scattering matrixes are

in good agreement with the database from Yang et al. (2013). The scattering matrix of the warm cloud was computed according to the Mie theory for water spheres having the gamma size distribution with $d_{eff} = 18.0$ μm (the standard deviation of 5.3 μm). The scattering matrix of the coarse-aerosol was simulated according to the work by Dubovik et al. (2006) as the "Mixture 1" of spheroids with different values of the axis ratio and the log-normal size distribution (the mean radius of 2 μm, the standard deviation of 0.6 μm, $d_{eff} = 4.75$ μm). To underline the differences in scattering properties, we show, as an example, the

normalized phase functions $p(\theta)$ for the wavelengths of 0.532 μm in Fig. 1 ($\theta$ is the scattering angle); their behaviour at forward and backward angles can be seen in the insets.

For subsequent discussions, we give in Table 1 parameters that have a significant place in multiple-scattering theory. The effective diameter is usually used to estimate the Fraunhofer diffraction angle $\theta_d = \lambda/d_{eff}$. The asymmetry parameter $g$, i.e. the first moment of a phase function, is one of basic parameters of the radiative transfer theory. $\theta_{max}$ is the angle where the

function $p(\theta) \cdot sin(\theta)$ takes the maximum. The value of the asymmetry parameter $g$ of the coarse aerosol is quite close to the values observed in volcanic degassing plumes (Shcherbakov et al., 2016) and Sahara dust aerosols (Horvath et al., 2018). The values of $g$ for the water cloud and the cirrus cloud are in good agreement with the experimental data of the work by Jourdan et al. (2010).

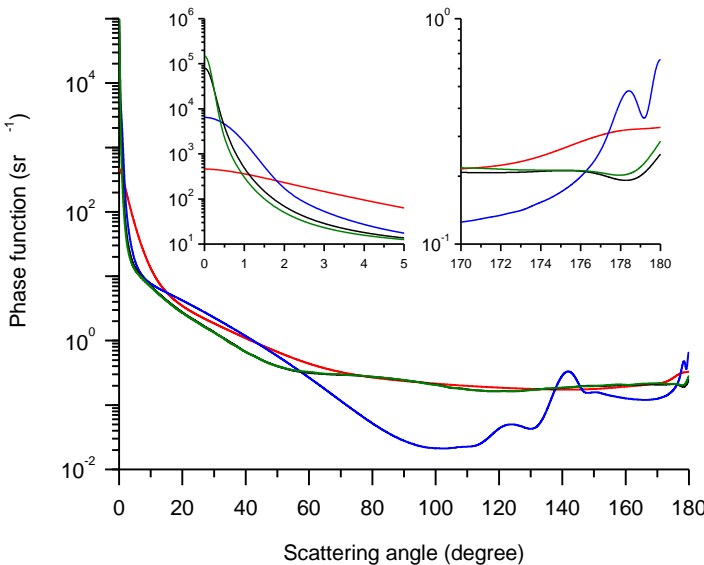

**Figure 1.** Normalized phase functions: coarse-aerosol – red lines, water cloud – blue lines, JS cirrus – black lines, Ci cirrus – green lines.

**Table 1.** Integral parameters of the phase functions.

|  | Coarse aerosol | Water cloud | JS cirrus | Ci cirrus |
|---|---|---|---|---|
| Effective diameter (µm) | 4.75 | 18.0 | 56.8 | 80.0 |
| Wavelength 0.355 µm | | | | |
| $\theta_d$ (degree) | 4,28 | 1,13 | 0,72 | 0,25 |
| $\theta_{max}$ (degree) | 1,41 | 0,41 | 0,11 | 0,08 |
| Asymmetry parameter | 0,77 | 0,87 | 0,76 | 0,76 |
| Wavelength 0.532 µm | | | | |
| $\theta_d$ (degree) | 6,42 | 1,69 | 1,07 | 0,38 |
| $\theta_{max}$ (degree) | 2,14 | 0,62 | 0,16 | 0,13 |
| Asymmetry parameter | 0,74 | 0,87 | 0,77 | 0,77 |

## 3 Background and basic properties of multiple scattering impact

We use the following notations in this work. The function $S_1(h)$ characterizes lidar signals in the single-scattering approximation (corrected for the offset and instrumental factors):



$$S_1(h) = [\beta_p(h) + \beta_m(h)] \cdot T^2(h), \tag{1}$$

where $h$ is the distance from the lidar (the altitude in the case of the ground-based zenith-looking lidar); $\beta_p(h)$ and $\beta_m(h)$

represent the backscatter contributions from particles and from the atmospheric molecules; $T^2(h) = T_m^2(h) \cdot T_p^2(h)$ is the two-way transmittance from the lidar to the range $h$; $T_m^2(h)$ and $T_p^2(h)$ are the molecular and the particulate transmittances, respectively. $T_p^2(h) = 1$ if $h \leq h_b$, where $h_b$ is the distance to the cloud near end; $T_p^2(h) = \exp[-2\tau_p(h_b, h)]$ when $h \geq h_b$, where $\tau_p(h_b, h) = \int_{h_b}^{h} \varepsilon_p(h') dh'$ is the cloud optical depth, $\varepsilon_p(h)$ is the extinction coefficient of particles.

The term "apparent attenuated backscatter" (see, e.g., Chepfer et al., 1999) is employed for lidar signals $S_{MS}(h)$ computed in

multiple-scattering conditions (corrected for the offset and instrumental factors). Without loss of generality, we can assume that

$$S_{MS}(h) = G_{MS}(h) \cdot [\beta_p(h) + \beta_m(h)] \cdot T_m^2(h) \cdot T_p^2(h), \tag{2}$$

where the multiple scattering function (MSF) $G_{MS}(h)$ is the ratio

$$G_{MS}(h) = \frac{S_{MS}(h)}{S_1(h)}. \tag{3}$$

It is employed as a factor that corrects the lidar signal of the single scattering approximation. Such an approach was used in the automated algorithm of the Atmospheric Radiation Measurement Program's Raman lidar (Thorsen and Fu, 2015). As a matter of fact, Eq. (2) is no more than a mathematical expression that provides an easy way to assign the relationship between $S_{MS}(h)$ and $S_1(h)$.

A specific model of multiple scattering appears only when $G_{MS}(h)$ is given in an explicit form. The specific model that widely

used by the lidar community is based on the works by Platt (1973, 1979). It was proposed to account for "secondary scattering or higher order processes" using the factor (in our notations $\eta_{MS}(h)$) that "multiplies the optical depth, has a value less than unity and it may vary with altitude" (Platt, 1979). In that case, lidar signal that have been corrected for the offset and instrumental factors can be written as (Winker, 2003):

$$S_{MS}(h) = [\beta_p(h) + \beta_m(h)] \cdot T_m^2(h) \cdot \exp[-2\eta_{MS}(h)\tau_p(h_b, h)]. \tag{4}$$

It is a straightforward matter to transform Eq. (4) to the following form

$$S_{MS}(h) = [\beta_p(h) + \beta_m(h)] \cdot T_m^2(h) \cdot T_p^2(h) \cdot \exp\{2[1 - \eta_{MS}(h)]\tau_p(h_b, h)\}, \tag{5}$$

and obtain the relationship

$$G_{MS}(h) = \exp\{2[1 - \eta_{MS}(h)]\tau_p(h_b, h)\}. \tag{6}$$

Equations (3) and (6) lead directly to the well-known formula for the Multiple Scattering Function (MSF) (Winker, 2003)

$$\eta_{MS}(h) = 1 - \frac{1}{2 \cdot \tau_p(h_b, h)} \cdot \ln\left[\frac{S_{MS}(h)}{S_1(h)}\right], \tag{7}$$

which can be rewritten as

$$\eta_{MS}(h) = 1 - \frac{1}{2 \cdot \tau_p(h_b, h)} \cdot \ln[G_{MS}(h)]. \tag{8}$$

The relationships between $G_{MS}(h)$ and two types of MSFs can be found in Appendix A.





The interpretation of MSF $\eta_{MS}(h)$ plots should be done with appropriate caution because of the logarithm in the right-hand

side of Eqs. (7) – (8). If it is assumed that $\eta_{MS}(h) = const$, the impact $G_{MS}(h)$ of multiple scattering has an exponential growth rate as a function of the in-cloud optical depth $\tau_p(h_b, h)$ (see, Eqs. 3 and 6). If $G_{MS}(h)$ increases, but with a rate lower than exponential, $\eta_{MS}(h)$ increases. That feature is of importance for a complete understanding of the MSF $\eta_{MS}(h)$ behaviour at the near end of a particles layer. If $G_{MS}(h)$ has a faster than exponential growth rate, the MSF $\eta_{MS}(h)$ decreases.

Our Monte-Carlo simulations provide the range-dependent lidar signals in the single, the double, and the multiple scattering

conditions, that is, $S_1(h)$, $S_2(h)$, and $S_{MS}(h)$ with the spatial resolution of 20 m. The ratio, that is, the relative contribution of multiple scattering

$$R_{MSto1}(h) = [S_{MS}(h) - S_1(h)]/S_1(h) = G_{MS}(h) - 1 \qquad (9)$$

can be computed directly from the MC data; $R_{MSto1}(h) = 0$ if $h \leq h_b$. We recall that $R_{MSto1}(h)$ are largely used in the literature to address effects of multiple scattering on lidar signals when direct problems are dealt with.

It is instructive to see the double scattering impact especially when the multiple scattering effect is not high. Thus, we will use the notations $G_2(h)$, $\eta_2(h)$, and $R_{2to1}(h)$ that are computed according Eqs. (3), (7), and (9), respectively, with the difference that $S_{MS}(h)$ is replaced by $S_2(h)$. MC simulations never provide continuous functions. Quantization of lidar data, in our case it means the integration over a distance interval, that is, a range gate is always required. It is of importance that the MSFs $\eta_2(h_i)$ and $\eta_{MS}(h_i)$ are computed with $h_i$ assigned to the middle of the range gate when Eq. (7) is used (see details in Appendix

B).

Some important features of multiple-scattering effect can be revealed when lidar signals are simulated within a quite large range of the optical depth in spite of limitations imposed by technical characteristics of receivers. Figure 2 shows results of two cases that are quite distinguished in terms of the configuration and particles properties. The both simulations were performed with the spatial resolution of 20 m, the total number of photons of $4 \cdot 10^{10}$; 50 orders of scattering were taken into

account.

We used for the first case (Figs. 2a and 2b) the configuration of the MUSCLE (MUltiple Scattering in Lidar Experiments) community (Bissonnette et al., 1995; Winker and Poole, 1995). The distance to the water cloud C1 is low ($h_b = 1$ km); the lidar transmitter has the wavelength of 1.064 μm and the divergence of 0.1 mrad (full angle); the full RFOV is of 1.0 mrad; the particles extinction coefficient is large (of 17.25 km⁻¹), which is favourable for multiple scattering. In the work by

Bissonnette et al. (1995) the results are shown for the penetration depth up to 300 m ($\tau_p(h_b, h) = 5.175$). Within that range, the McRALI simulations are in total agreement with the MUSCLE data (see details in Alkasem et al. (2017)). In this work, we show our MC data for the penetration depth up to 1.15 km ($\tau_p(h_b, h) = 19.8$).





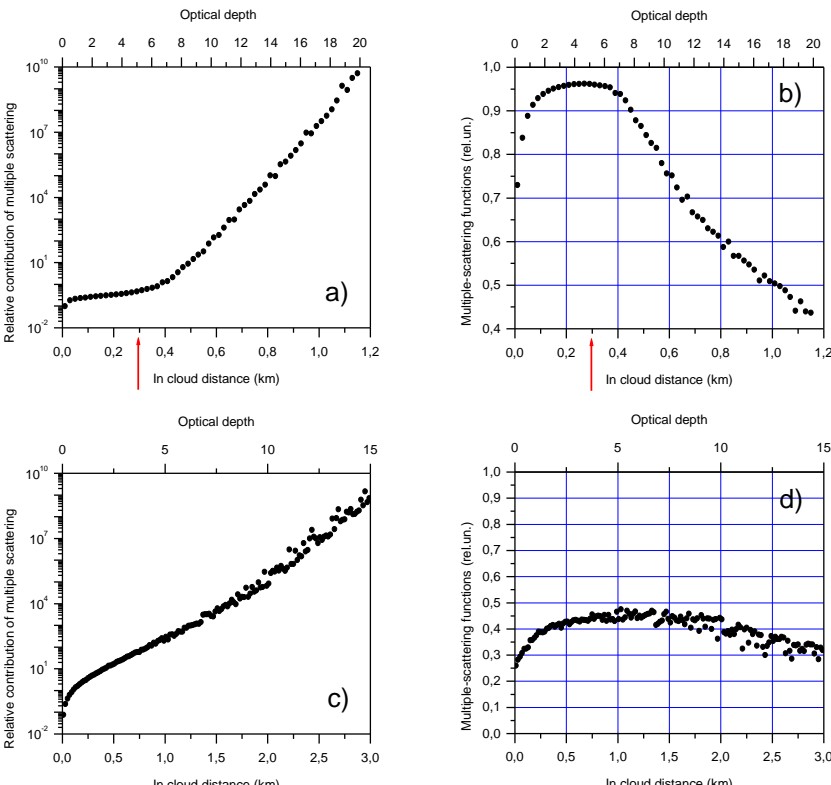

**Figure 2.** Multiple scattering contributions $R_{MSto1}$ to lidar signals (a, c) and multiple scattering functions $\eta_{MS}(d)$ (b, d). MUSCLE case – (a – b), CALIOP case – (c – d). The red arrow indicates the far end of the MUSCLE-comparison (Bissonnette et al, 1995).

The second case (Figs. 2c and 2d) deals with the configuration of the Cloud-Aerosol Lidar with Orthogonal Polarization (CALIOP) (Winker et al., 2010). The nadir-looking lidar is at the altitude of 705 km; the transmitter has the wavelength of 0.532 μm and the divergence of 0.11 mrad (full angle); the full RFOV is of 0.13 mrad. A cirrus cloud (Ci) has the extinction coefficient of 5.0 km⁻¹.

To evidence the multiple-scattering effect, two functions are mostly used in the literature, namely, the relative contribution of multiple scattering $R_{MSto1}(h)$ and the MSF $\eta_{MS}(h)$. The both function are shown in Fig. 2; the red arrow indicates the far end of the MUSCLE-comparison (Bissonnette et al, 1995). The literature and our wealth of experience in lidar-signals MC-simulations suggest that the function $R_{MSto1}(h)$ possesses at the near end of a cloud the features, which are common for most if not all configurations and particles properties. In the beginning, $R_{MSto1}(h)$ is linearly proportional to the in-cloud distance (see Figs. 2a and 2c, and Sections below). Then, the curve bends to the right at the in-cloud distance $d_1$ ($d_1 \approx 0.03$ km in Fig.





2a and $d_1 \approx 0.07$ km in Fig. 2c). It remains increasing with the same rate within a quite large range. The curve bends upward
at the in-cloud distance $d_2$ ($d_2 \approx 0.52$ km in Fig. 2a and $d_2 \approx 1.35$ km in Fig. 2c), i.e. at the optical depth somewhat between
6.0 and 7.0. The differences in lidar configurations and/or particles properties result in the width of the intervals $[0, d_1]$ and
$[d_1, d_2]$ as well as in the increasing rate of $R_{MSto1}(h)$ within these intervals. We hypothesize that the multiple scattering effect
within the range $[0, d_2]$ is mostly due to the photons that remain within or close to the RFOV; to the contrary, the photons that
walk a lot outside the RFOV become dominant when $d > d_2$. (We recall that $\tau_p(h_b, h) \approx 7.0$ is at the bound of technical
capacities of contemporary lidars.)

The functions $R_{MSto1}(h)$ and $\eta_{MS}(h)$ have the direct relationship (see Eqs. (8) – (9)). At the same time, the MSF $\eta_{MS}(h)$ (Figs.
2b and 2d) can provide a somewhat more keen insight into effects of multiple scattering. For example, it is seen in Fig. 2b the
pronounced change in the $\eta_{MS}(h)$ behaviour at the optical depth $\tau_p(h_b, h) \approx 7.0$, which implies that some other physical
events become dominant. In Fig. 2d that property can be observed even though it is less pronounced. All that leads to the
conclusion that our empirical model (see below) is limited to the cases when the optical depth $\tau_p(h_b, h) < 7.0$.

## 4 Empirical model

It follows from our Monte Carlo simulations for different configurations and/or particles properties that the computed functions
$G_{MS}(h)$ show the similar behaviour within the range $[0, d]$ of the in-cloud distance $d$ when plotted as a log–linear graph.
(Typical examples can be seen in Figs. 3, 5, and 8.) That similarity led us to the following empirical model:

$$G_{MS}(d) = \exp[V(d, \boldsymbol{a})], \tag{10}$$

where the function $V(d, \boldsymbol{a})$ has only three free parameters $\boldsymbol{a} = \{a_1, a_2, a_3\}$ and the domain of definition $d \geq 0$:

$$V(d, \boldsymbol{a}) = a_1 \cdot \arctan(a_2 \cdot d) + a_3 \cdot d. \tag{11}$$

If values of $V(d, \boldsymbol{a})$ are quite small, i.e., the impact of multiple scattering is quite low, we can write using the first two terms
of the expansion in powers of $V(d, \boldsymbol{a})$ of the exponential function:

$$G_{MS}(d) = \exp[V(d, \boldsymbol{a})] \approx 1 + V(d, \boldsymbol{a}). \tag{12}$$

Equations (9) and (12) lead to the relationship $R_{MSto1}(d) \approx V(d, \boldsymbol{a})$. Thus, the simplified version of the empirical model can
be used to fit simulations data $R_{MSto1}(d)$ with the function $V(d, \boldsymbol{a})$. Some properties of $V(d, \boldsymbol{a})$ can be easily deduced:

$$V(d, \boldsymbol{a}) \approx (a_1 \cdot a_2 + a_3) \cdot d = b \cdot d \tag{13}$$

at small values of $d$, i.e., at the cloud near-end $h_b$. In other words, $V(d, \boldsymbol{a})$ is linearly proportional to the in-cloud distance with
the coefficient $b = (a_1 \cdot a_2 + a_3)$ and $V(0, \boldsymbol{a}) = 0$.

At large values of $d$,

$$V(d, \boldsymbol{a}) \approx a_1 \cdot \frac{\pi}{2} + a_3 \cdot d, \tag{14}$$

that is, another time it is linearly proportional to the in-cloud distance but with the coefficient $a_3$.





The function $\exp[V(d, \boldsymbol{a})]$ has the following properties.

$$\exp[V(d, \boldsymbol{a})] \approx 1 + (a_1 \cdot a_2 + a_3) \cdot d \tag{15}$$

at small values of $d$. At large values of $d$,

$$\exp[V(d, \boldsymbol{a})] \approx \exp\left[a_1 \cdot \frac{\pi}{2} + a_3 \cdot d\right] = \exp\left[a_1 \cdot \frac{\pi}{2}\right] \cdot \exp[a_3 \cdot d], \tag{16}$$

that is, it increases exponentially with the in-cloud distance.

It is worthwhile to see how the MSF $\eta_{MS}(d)$ is expressed in terms of the empirical model. Equations (8, 10 – 11) lead to the following relationship when $\varepsilon_p = \text{const}$:

$$\eta_{MS}(d) = 1 - \frac{a_3}{2\cdot\varepsilon_p} - \frac{a_1}{2\cdot\varepsilon_p\cdot d} \cdot \arctan(a_2 \cdot d). \tag{17}$$

Using properties of the arctangent, we can write:

$$\eta_{MS}(d) \approx 1 - \frac{1}{2\cdot\varepsilon_p} \cdot (a_1 \cdot a_2 + a_3) + \frac{a_1\cdot a_2^3}{6\cdot\varepsilon_p} \cdot d^2 \tag{18}$$

at small values of the in-cloud distance $d$, and

$$\eta_{MS}(d) \approx 1 - \frac{a_3}{2\cdot\varepsilon_p} - \frac{\pi\cdot a_1}{4\cdot\varepsilon_p} \cdot \frac{1}{d} \tag{19}$$

at large values of $d$ within some range of $d < d_2$ (see Section 3).

There exists another function, which is somewhat similar to $\arctan(x)$, namely, the hyperbolic tangent $\tanh(x)$. The hyperbolic tangent is frequently used in the radiative transfer theory. Thus, we assayed another empirical model where $\arctan(x)$ was replaced by $\tanh(x)$. Our tests (not shown here) revealed that the model Eq. (10) always provides fitting errors lower or much lower than the model with the hyperbolic tangent.

In this work, all values of the fitting parameters $\boldsymbol{a} = \{a_1, a_2, a_3\}$ are given in tables of the supplementary material to avoid overloading the text.

**5 Low distances and small optical depths**

For brevity sake, we will use the term "usual operational conditions" (UOCs) when the distance from a lidar to a layer of particles is lower than 15 km, the RFOV ≤ 1 mrad, the emitter field-of-view (EFOV) ≤ 0.2 mrad, EFOV ≪ RFOV, the extinction coefficient $\varepsilon \leq 1$ km⁻¹. All simulations of this Section were performed for the UOCs and at the wavelength of 0.532 µm.

**5.1 Ground-based lidar**

Figure 3 and 4 shows the results of our MC simulations reported in terms of the ratios $R_{MSto1}$ and MSF $\eta_{MS}(d)$, respectively. The ground-based lidar is at the altitude $h=0$ km, the layer is within the altitude $h$ range of [8 – 11] km. The distance to the layer base is of 8 km. The number of photons emitted by the lidar was of $4 \cdot 10^{10}$. We use the same type of notations in both figures. The left hand column corresponds to the full RFOV of the lidar of 1.0 mrad; the right hand column corresponds to the





full RFOV of 0.25 mrad. Blue, red, green, and purple points show the MC simulation results obtained with the extinction

coefficient of 1.0 km$^{-1}$, 0.5 km$^{-1}$, 0.2 km$^{-1}$, and 0.06 km$^{-1}$, respectively. The black curves in Fig. 3 represent the fitting results; each curve corresponds to its own set of points $R_{MSto1}(d)$. The MC data were fitted by the $V(d, \boldsymbol{a})$ function, that is, we computed the values of the free parameters $\boldsymbol{a} = \{a_1, a_2, a_3\}$ using the ordinary least squares approach.

The cases of the low value $\varepsilon_p(h) = 0.06$ km$^{-1}$ have the following peculiarities. The double scattering dominates to the extent that higher scattering orders can be neglected. The effect of multiple scattering is very low for the coarse aerosol and the water

cloud (therefore, the corresponding data are not shown in this work). As for JS cirrus and Ci cirrus, the corresponding MC simulations were performed with the number of photons emitted by the lidar 10 times higher, i.e., of $4 \cdot 10^{11}$ in order to decrease random noise.

It is seen in Fig. 3 that the function $V(d, \boldsymbol{a})$ fits well the MC data, which vary widely in terms of values and curve shape. (The corresponding values of the parameters $\boldsymbol{a} = \{a_1, a_2, a_3\}$ are given in Tables S1 and S2 of the supplementary material.)

Despite large shape-variation of the ratios $R_{MSto1}$ in Fig. 3, all curve shapes are in total agreement with the literature. For example, the curves corresponding to the water and cirrus clouds look like curves in the following works (Fig. 5, Kunkel and Weinman, 1976; Fig. 3, Wandinger, 1998; Fig. 6, Eloranta, 1998). As for the ratios $R_{MSto1}(d)$ corresponding to the coarse aerosol, they are linearly proportional to the penetration depth $d$ starting from about $d = 250$ m. That feature is in agreement with Fig. 2 of the work by Ackermann et al. (1999).





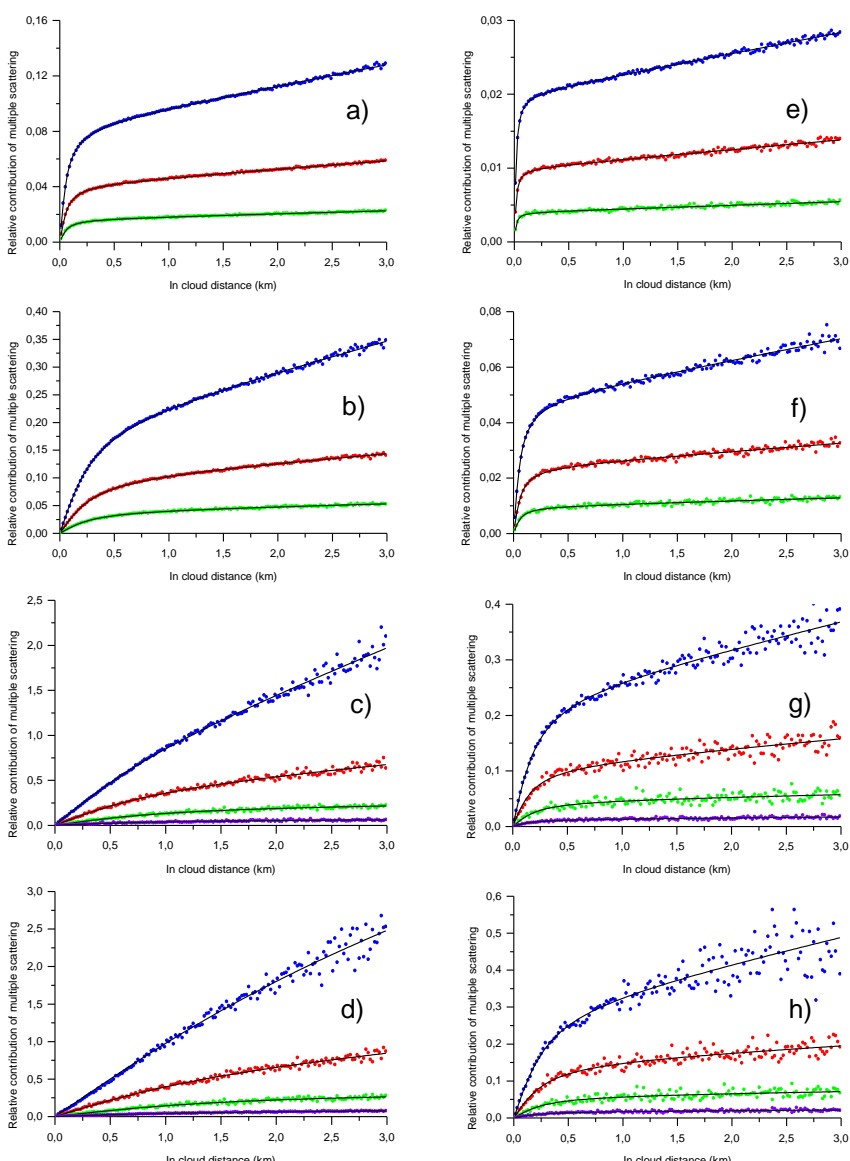


**Figure 3.** Multiple scattering contributions $R_{MSto1}$ to lidar signals. Points – MC simulations, black lines – fitting with the empirical model. The full RFOV is of 1.0 mrad (a – d) and of 0.25 mrad (e – h). Coarse-aerosol (a, e), water cloud (b, f), JS cirrus (c, g), Ci cirrus (d, h). The extinction coefficient is of 1.0 km$^{-1}$ (blue points), 0.5 km$^{-1}$ (red points), 0.2 km$^{-1}$ (green points), and 0.06 km$^{-1}$ (purple points). The distance to the cloud base is of 8 km.






The applicability of the single scattering approximation (SSA) to lidar signals can be assessed on the base of $R_{MSto1}(d)$ values. The percentage of the multiple-scattering relative contribution to lidar signals is shown in Tables 2 – 3. Those values were computed using corresponding $V(d, \boldsymbol{a})$ functions, which allow random-noise smoothing. We recall that random noise is inherent in MC simulations. The sample size of our modelling is already very large, i.e., at the limit of our computing capacities.

In the subsequent discussion, we assume the 5% threshold for the $R_{MSto1}(d)$ values to consider the single scattering approximation as acceptable. The values exceeding that threshold are highlighted by the red colour in Tables 2 – 3. In our opinion, the most important outcome is the fact that the SSA has to be rejected in the cases of cirrus clouds when the RFOV is of 1.0 mrad (see Table 2). Even with $\varepsilon_p = 0.06$ km$^{-1}$ and $d = 1.0$ km the multiple scattering contribution is about 4 %. As for the RFOV of 0.25 mrad (see Table 3), the SSA is acceptable for the cirrus clouds only with $\varepsilon_p = 0.1$ km$^{-1}$ or lower.

Actually, the overwhelming majority of the values in Table 2 is out of the threshold. Thus, the RFOV of 1.0 mrad cannot be recommended when the distance to a particles layer is about 8 km of higher. The SSA is acceptable for the coarse aerosol when the RFOV is of 0.25 mrad (see Table 3). That conclusion holds true for the fine-mode aerosols (they have lower values of the effective diameter). As for the water cloud, the SSA is acceptable when $\varepsilon_p \lesssim 0.5$ km$^{-1}$ and the RFOV is of 0.25 mrad.

As it was mentioned above, the majority of works take into account multiple scattering employing one of possible multiple-scattering functions (MSF). Moreover, the simplified version $\eta_{MS}(d) = $ const of the MSF $\eta_{MS}(d)$ is frequently employed in inverse problems. That is the reason why, we provide $\eta_{MS}(d)$ computed on the base of our MC simulations (see Eq. (7) and details in Appendix B).

**Table 2.** Multiple scattering contribution to lidar signals in percent of the single scattering. The distance to the cloud base is of 8 km; the lidar RFOV is of 1.0 mrad.

| Penetration depth | 1.0 km | | | | 3.0 km | | | |
|---|---|---|---|---|---|---|---|---|
| Extinction coefficient (km$^{-1}$) | Coarse aerosol | Water cloud | JS cirrus | Ci cirrus | Coarse aerosol | Water cloud | JS cirrus | Ci cirrus |
| 0.06 | | | 3.9 | 4.2 | | | **6.2** | **7.7** |
| 0.20 | 1.8 | 4.0 | **13.3** | **14.4** | 2.3 | **5.3** | **21.7** | **26.3** |
| 0.50 | 4.6 | **10.2** | **35.7** | **39.6** | **5.9** | **14.3** | **67.5** | **84.5** |
| 1.00 | **9.6** | **22.2** | **84.9** | **96.9** | **12.8** | **34.7** | **197.6** | **246.8** |



**Table 3.** Multiple scattering contribution to lidar signals in percent of the single scattering. The distance to the cloud base is of 8 km; the lidar RFOV is of 0.25 mrad.

| Penetration depth | 1.0 km | | | | 3.0 km | | | |
|---|---|---|---|---|---|---|---|---|
| Extinction coefficient (km$^{-1}$) | Coarse aerosol | Water cloud | JS cirrus | Ci cirrus | Coarse aerosol | Water cloud | JS cirrus | Ci cirrus |
| 0.06 | | | 1.3 | 1.7 | | | 1.7 | 2.2 |
| 0.20 | 0.4 | 1.0 | 4.5 | **5.6** | 0.5 | 1.3 | **5.7** | **7.1** |
| 0.50 | 1.1 | 2.6 | **11.6** | **14.7** | 1.4 | 3.3 | **15.8** | **19.5** |
| 1.00 | 2.3 | **5.4** | **25.7** | **32.3** | 2.8 | **7.0** | **36.8** | **48.7** |

As for the MSF $\eta_{MS}(d)$ in Fig. 4, the values of $\eta_{MS}(d)$ are so close within each panel that the green points ($\varepsilon_p(d) = 0.2$ km$^{-1}$) sometimes totally cover other colours. In other words, the impact of multiple scattering has the same growth rate when plotted as a function of the in-cloud depth and the extinction coefficient is quite low $\varepsilon_p(d) \leq 1.0$ km$^{-1}$. Each type of particles, of course, has its own growth rate. That property is not valid when the impact of multiple scattering is quite high (see Section 6 below).

Generally, there is much in common between all curves $\eta_{MS}(d)$. Moreover, such kind of curves can be found in the literature. For example, the MSF $\eta_{MS}(d)$ of the cirrus clouds in Figs. 4(g – h) are closely similar to the curves in Fig. 14 of the work by Platt (1981); the discrepancy between values is most likely due to difference in phase function properties within the forward diffraction peak.

It is seen in Fig. 4 that $\eta_{MS}(d)$ is a nonlinear function. In our opinion, $\eta_{MS}(d) = $ const is a rough approximation while lidar signals are recorded in the usual operational conditions. Its only justification is that it is easily adapted to a solution of an inverse problem. Generally, the solution should be biased and the level of consequent errors depends on a specific algorithm used to solve the inverse problem. A study of biases can be performed using the results of this work, i.e., the $V(d, \boldsymbol{a})$ function along with the values of the parameters $\boldsymbol{a}$.

As expected, our simulations confirm general properties of multiple scattering effect on lidar signals that can be found in the literature (see, e.g., Eloranta, 1998). Namely, the effect of multiple scattering as well as the relative contribution of the third and higher orders of scattering increase with increased extinction coefficient, in-cloud distance and receiver field-of-view. The proportion of light scattered within very small angles, that is, within the forward diffraction peak (see the inset in Fig. 1), is of upmost importance. That proportion is characterized by the angular width $\theta_d$ of the diffraction peak in the work by Eloranta, (1998). To the contrary, the asymmetry parameter is of little significance for multiple-scattering effects on lidar signals





recorded in the UOCs. For instance, the asymmetry-parameter values of the coarse aerosol and the cirrus clouds differ little (see Table 1), whereas there are fundamental differences in the multiple scattering.

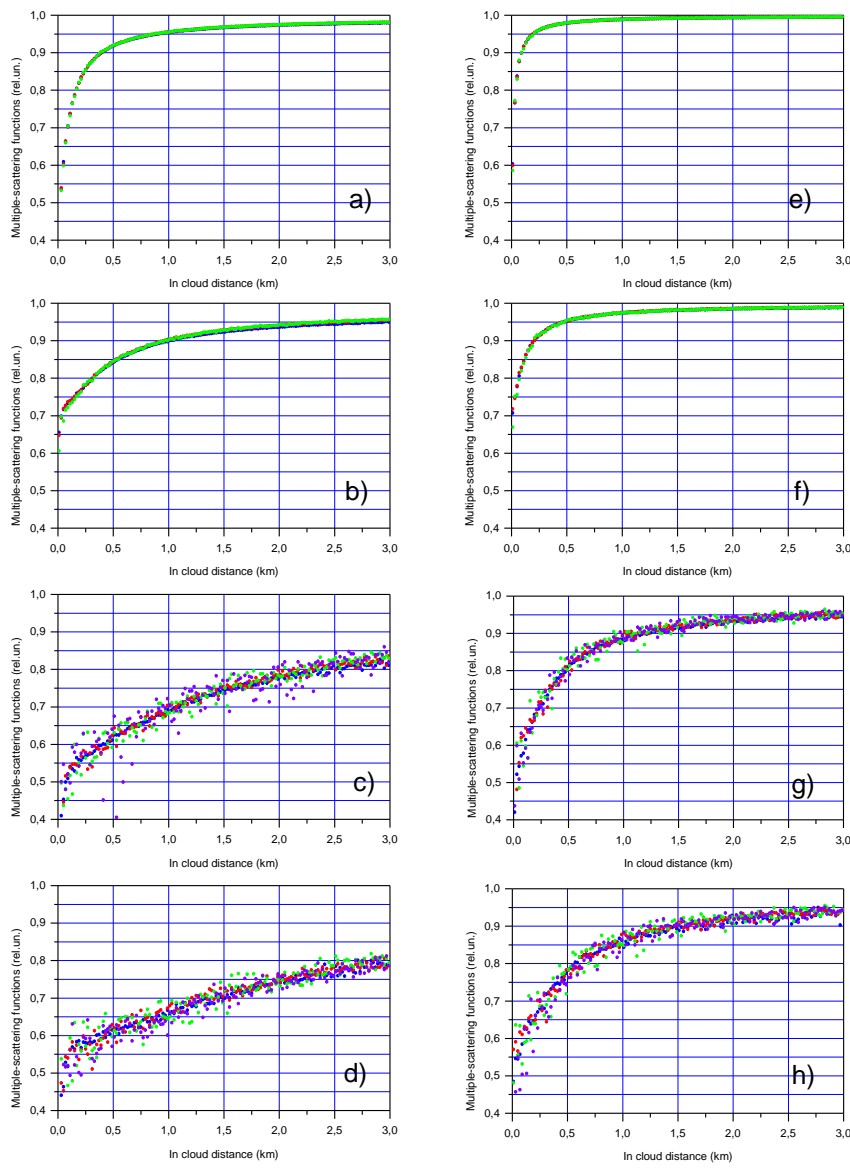


**Figure 4.** MC simulations of multiple scattering functions $\eta_{MS}(d)$. The full RFOV is of 1.0 mrad (a – d) and of 0.25 mrad (e – h). Coarse-aerosol (a, e), water cloud (b, f), JS cirrus (c, g), Ci cirrus (d, h). The extinction coefficient is of 1.0 km$^{-1}$ (blue points), 0.5 km$^{-1}$ (red points), 0.2 km$^{-1}$ (green points), and 0.06 km$^{-1}$ (purple points). The distance to the cloud base is of 8 km.



It is seen that the ratios $R_{MSto1}(d)$ and the MSFs $\eta_{MS}(d)$ of the jet-stream cirrus and the Ci cirrus are quite close, and so are

the values of $\theta_{max}$. The difference in the values of $\theta_d$ is much larger. In our opinion, the angle $\theta_{max}$ is more appropriate for

use as one of the parameters that govern the effect of multiple scattering on lidar signals.

The same kind of study was done in view of the double scattering contribution. The main conclusion is that the empirical

model fits well the functions $R_{2to1}(d)$. A representative example can be seen in Fig. S1 of the supplementary material.


**5.2 Airborne lidar**

All but one simulation conditions, that is, the input data from the foregoing subsection were used in our MC simulations for

the airborne lidar. Namely, we are dealing with the coaxial zenith-looking lidar that is at the altitude of 7 km and the distance

to the cloud base is of 1 km. We recall that the results of Section 5 are presented so that they remain unaltered when the lidar

pointing angle and/or the layer altitude vary provided that the distance to the cloud base/border remains unchanged.

As in Subsection 5.1, the results of our MC simulations are reported in terms of the ratios $R_{MSto1}(d)$ and MSF $\eta_{MS}(d)$ (Figs.

5 and 6, respectively). Another time, the same type of notations was used in both figures. The left hand column corresponds

to the full RFOV of the lidar of 1.0 mrad; the right hand column corresponds to the full RFOV of 0.25 mrad. Blue, red, green,

and purple points show the MC simulation results obtained with the extinction coefficient of 1.0 km[-1], 0.5 km[-1], 0.2 km[-1], and

0.06 km[-1], respectively. The black curves in Fig. 5 represent the fitting results; each curve corresponds to its own set of points

$R_{MSto1}(d)$. (The corresponding values of the parameters $\boldsymbol{a} = \{a_1, a_2, a_3\}$ are given in Tables S3 and S4 of the supplementary

material.) The effect of multiple scattering is much lower when compared to the ground-based lidar. That does not affect the

quality of fitting with the $V(d, \boldsymbol{a})$ function.

**Table 4.** Multiple scattering contribution to lidar signals in percent of the single scattering. The distance to the cloud base is

of 1 km; the lidar RFOV is of 1.0 mrad.

| Penetration depth | 1.0 km | | | | 3.0 km | | | |
|---|---|---|---|---|---|---|---|---|
| Extinction coefficient (km[-1]) | Coarse aerosol | Water cloud | JS cirrus | Ci cirrus | Coarse aerosol | Water cloud | JS cirrus | Ci cirrus |
| 0.06 | | | 1.2 | 1.4 | | | 2.3 | 2.8 |
| 0.20 | 0.4 | 0.9 | 3.8 | 4.7 | 0.8 | 1.9 | **8.0** | **9.6** |
| 0.50 | 1.0 | 2.3 | **9.8** | **12.1** | 2.1 | 4.8 | **21.3** | **26.4** |
| 1.00 | 2.0 | 4.8 | **21.2** | **26.1** | 4.2 | **10.3** | **50.4** | **65.6** |



**Table 5.** Multiple scattering contribution to lidar signals in percent of the single scattering. The distance to the cloud base is of 1 km; the lidar RFOV is of 0.25 mrad.

| Penetration depth | 1.0 km | | | | 3.0 km | | | |
|---|---|---|---|---|---|---|---|---|
| Extinction coefficient $(km^{-1})$ | Coarse aerosol | Water cloud | JS cirrus | Ci cirrus | Coarse aerosol | Water cloud | JS cirrus | Ci cirrus |
| 0.06 | | | 0.3 | 0.4 | | | 0.6 | 0.7 |
| 0.20 | 0.1 | 0.2 | 1.0 | 1.2 | 0.2 | 0.5 | 2.1 | 2.5 |
| 0.50 | 0.2 | 0.6 | 2.6 | 3.3 | 0.5 | 1.2 | **5.3** | **6.6** |
| 1.00 | 0.5 | 1.1 | **5.3** | **6.7** | 1.0 | 2.4 | **11.1** | **14.6** |


The percentage of the multiple-scattering relative contribution to lidar signals is shown in Tables 4 – 5. Those values were computed using corresponding $V(d, \boldsymbol{a})$ functions, which allow random-noise smoothing. As expected, the multiple-scattering contribution decreased with the distance to the cloud base decreased. More specifically, the ratio $R_{MSto1}$ decreased from 2.7 to 4.8 times when the distance to the cloud base decreased by the factor of eight. The reduction is more significant for the

penetration depth of 1 km. It is around 4.5 times for the full RFOV of 0.25 mrad. When the full RFOV is of 1.0 mrad, the reduction is around 4.5 times for the coarse aerosol and the water cloud, and around 3.5 times for the cirrus clouds. The reduction is clearly lower, i.e. around 3 times for the penetration depth of 3 km.

Again, we assume the 5% threshold for the $R_{MSto1}(d)$ values to consider the single scattering approximation as acceptable. The values exceeding that threshold are highlighted by the red colour in Tables 4 – 5. Special attention should be given to the

fact that in the cases of the cirrus clouds the effect of multiple scattering is at levels below the threshold only at quite low values of the extinction coefficient.

Again, we provide $\eta_{MS}(d)$ computed on the base of our MC simulations (see Eq. (7) and details in Appendix B). We can conclude another time that $\eta_{MS}(d)$ is a nonlinear function, and $\eta_{MS}(d) = $ const is a rough approximation while lidar signals are recorded in the usual operational conditions.

The general features of the ratios $R_{MSto1}(d)$ discussed at the end of in Section 5.1 hold true for the airborne lidar.





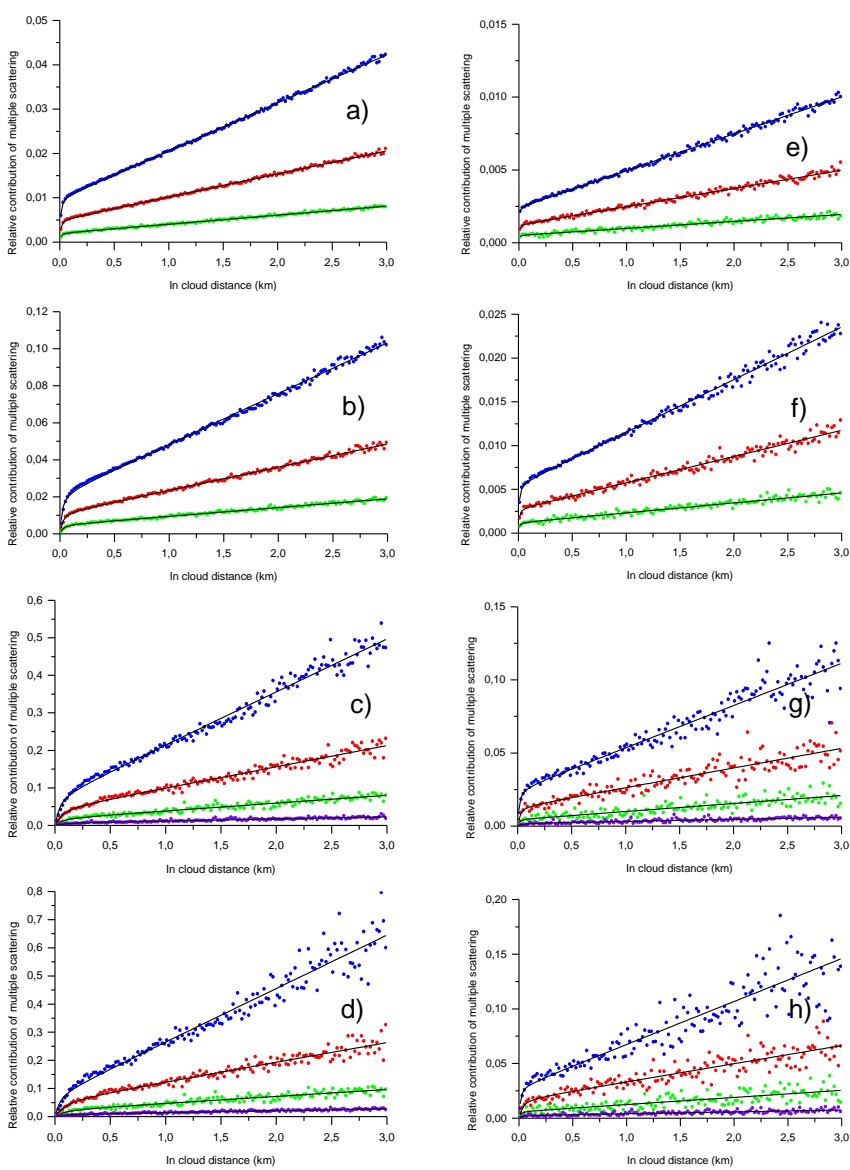

**Figure 5.** Multiple scattering contributions $R_{MSto1}$ to lidar signals. Points – MC simulations, black lines – fitting with the
empirical model. The full RFOV is of 1.0 mrad (a – d) and of 0.25 mrad (e – h). Coarse-aerosol (a, e), water cloud (b, f), JS
cirrus (c, g), Ci cirrus (d, h). The extinction coefficient is of 1.0 km$^{-1}$ (blue points), 0.5 km$^{-1}$ (red points), 0.2 km$^{-1}$ (green
points), and 0.06 km$^{-1}$ (purple points). The distance to the cloud base is of 1 km.



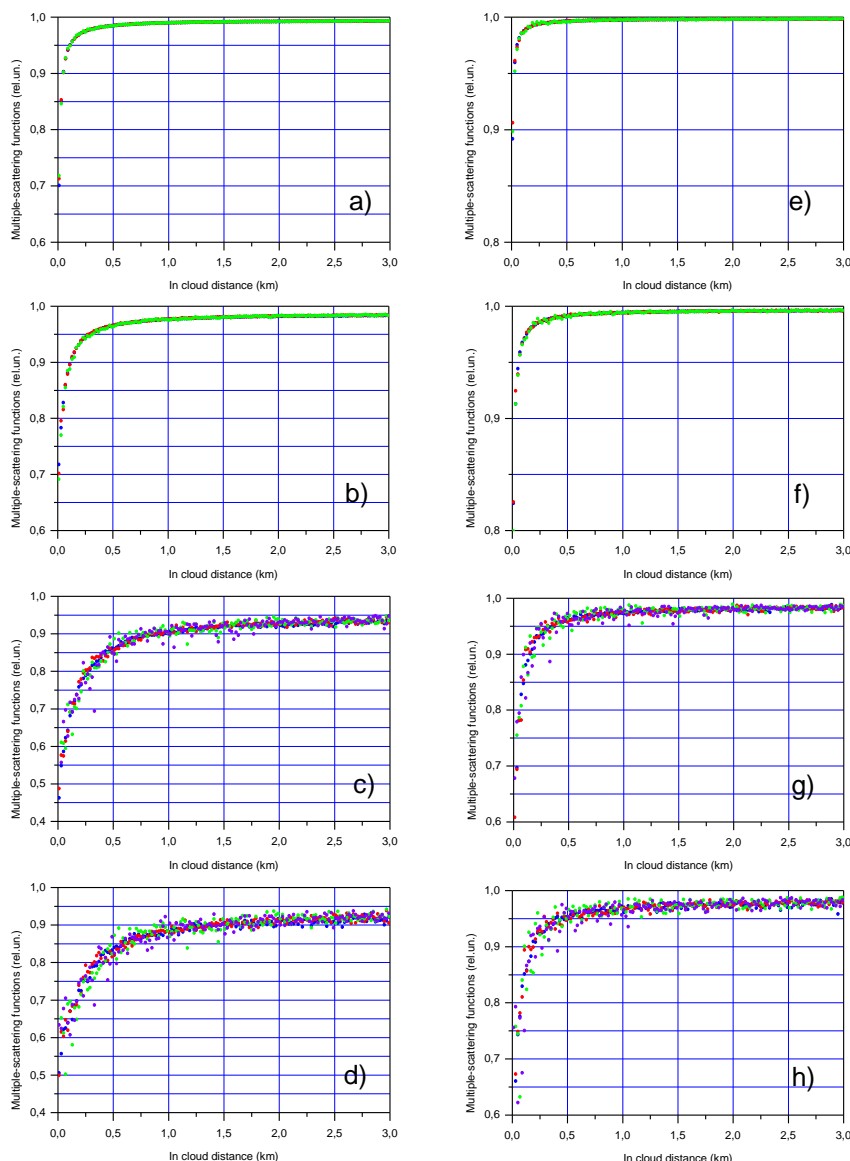

**Figure 6.** MC simulations of multiple scattering functions $\eta_{MS}(d)$. The full RFOV is of 1.0 mrad (a – d) and of 0.25 mrad (e – h). Coarse-aerosol (a, e), water cloud (b, f), JS cirrus (c, g), Ci cirrus (d, h). The extinction coefficient is of 1.0 km$^{-1}$ (blue points), 0.5 km$^{-1}$ (red points), 0.2 km$^{-1}$ (green points), and 0.06 km$^{-1}$ (purple points). The distance to the cloud base is of 1 km.





## 6 High impact of multiple scattering

### 6.1 Spaceborne lidars

#### 6.1.1 Moderate and small extinction coefficient

Figures 7 and 8 show examples of the multiple-scattering effect on signals of spaceborne lidars, i.e., the ratios $R_{MSto1}(d)$ and
the MSF $\eta_{MS}(d)$, respectively. The MS impact is high; accordingly, we use log–linear graphs in Fig. 7. As previously, we
maintain the same type of notations in both figures. The left hand column corresponds to the CALIOP configuration; the right
hand column corresponds to the ATLID configuration. Blue, red, green, and purple points show the MC simulation results
obtained with the extinction coefficient of 1.0 km$^{-1}$, 0.5 km$^{-1}$, 0.2 km$^{-1}$, and 0.06 km$^{-1}$, respectively. The black curves represent
the fitting results; each curve corresponds to its own set of points $R_{MSto1}(d)$.

The features of all ratios $R_{MSto1}(d)$ in Fig. 7 have much in common despite the large differences in scattering matrixes of
particles. Such kind of figures can be found in the literature (see, e.g., Fig 2, Winker, 2003). Moreover, we performed MC
simulations in the conditions of Fig. 4 (the second panel of the lower row) of the work by Wang et al. (2021). That is, a water
cloud has the extinction coefficient of 13.33 km$^{-1}$, the lidar transmitter has the wavelength of 0.532 μm, the full RFOV is of
0.10 mrad, the distance from the lidar to the cloud is of 703.7 km. The points of our simulated ratio $R_{MSto1}(d)$ superimpose
almost perfectly on the corresponding curve of the work by Wang et al. (2021). The MSF $\eta_{MS}(d)$ of the coarse aerosol in Figs.
7a and 7e resemble the curves in Fig. 3 of the work by Winker (2003). The difference in the values can be due to the fact that
the phase function of this work (see Fig. 1) has more pronounced forward scattering. There exists the discordance at the near
end of cirrus clouds between the MSF $\eta_{MS}(d)$ in Figs. 8c – 8d and Fig. 7 of the work by Winker (2003). It seems that the
discordance results from the 5 times finer spatial resolution used in our MC simulations.

It is seen that our empirical model Eq. (10) fits well the MC data in Fig. 7. The values of the free parameters $\boldsymbol{a} = \{a_1, a_2, a_3\}$
were computed using the ordinary least squares method. (The corresponding values of the parameters $\boldsymbol{a} = \{a_1, a_2, a_3\}$ are
given in Tables S5 and S6 of the supplementary material.) The red dash line indicates the 5% threshold for the multiple
scattering relative contribution $R_{MSto1}(d)$. In our opinion, the contribution below 5% is so exceptional that the single scattering
approximation should be never applied to data of spaceborne lidars.

As it is expected, the MSFs $\eta_{MS}(d)$ in the panels of Fig. 8 have lower values compared to the corresponding panels in Fig. 4.
It confirms that the MS effect is much more pronounced through all in-cloud range. Another time we can underline that
$\eta_{MS}(d) = \text{const}$ is only a rough approximation. At the same time, the approximation Eq. (19) seems to be valid within a quite
large range of the penetration depth. In addition, there are changes in MSFs behaviour. Unlike in the cases of Section 5, $\eta_{MS}(d)$
depends on the extinction coefficient (see, e.g., Figs. 9 a – b).

The general features of the ratios $R_{MSto1}(d)$ discussed at the end of in Section 5.1 hold true for spaceborne lidars. In particular,
the angle $\theta_{max}$ is more appropriate for use as one of the parameters that govern the effect of multiple scattering on lidar signals.





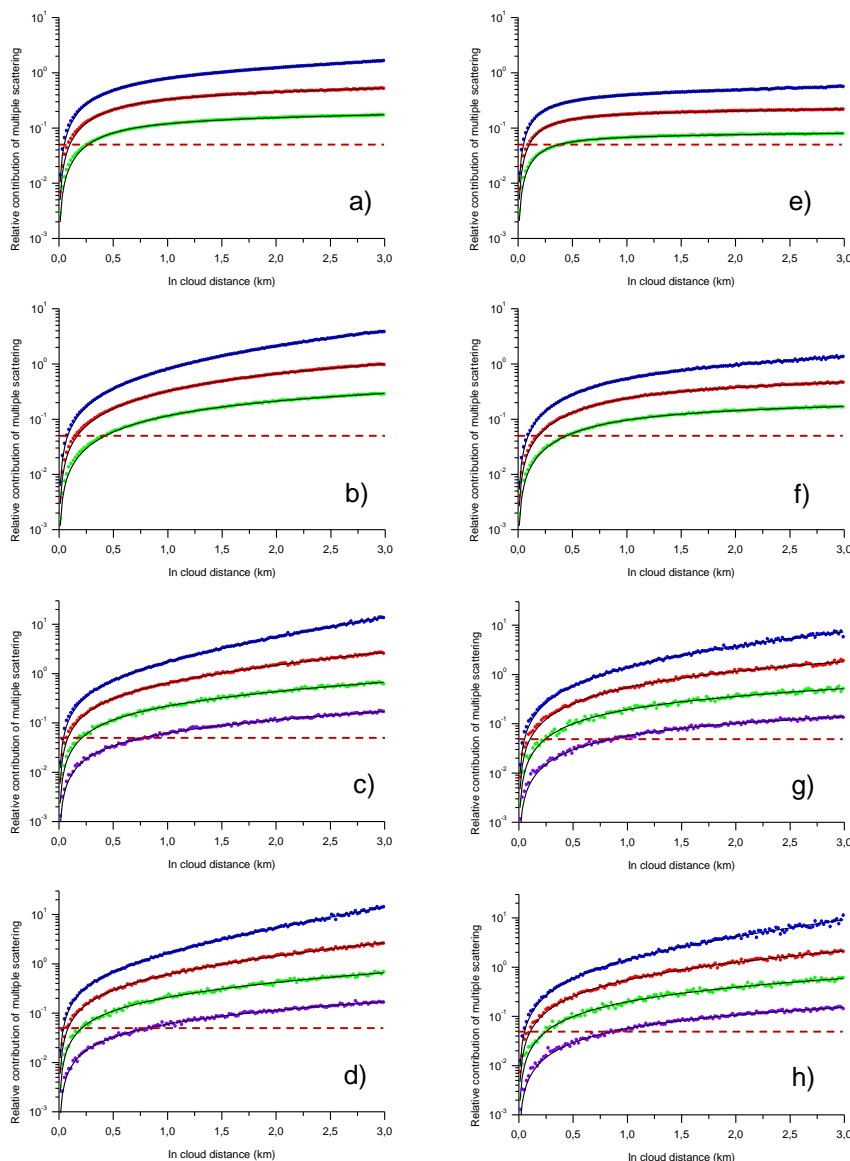

**Figure 7.** Multiple scattering contributions $R_{MSto1}$ to lidar signals. Points – MC simulations, black lines – fitting with the empirical model. The CALIOP configuration (a – d) and the ATLID configuration (e – h). Coarse-aerosol (a, e), water cloud (b, f), JS cirrus (c, g), Ci cirrus (d, h). The extinction coefficient is of 1.0 km$^{-1}$ (blue points), 0.5 km$^{-1}$ (red points), 0.2 km$^{-1}$ (green points), and 0.06 km$^{-1}$ (purple points). The red dash lines indicate the 5% threshold.





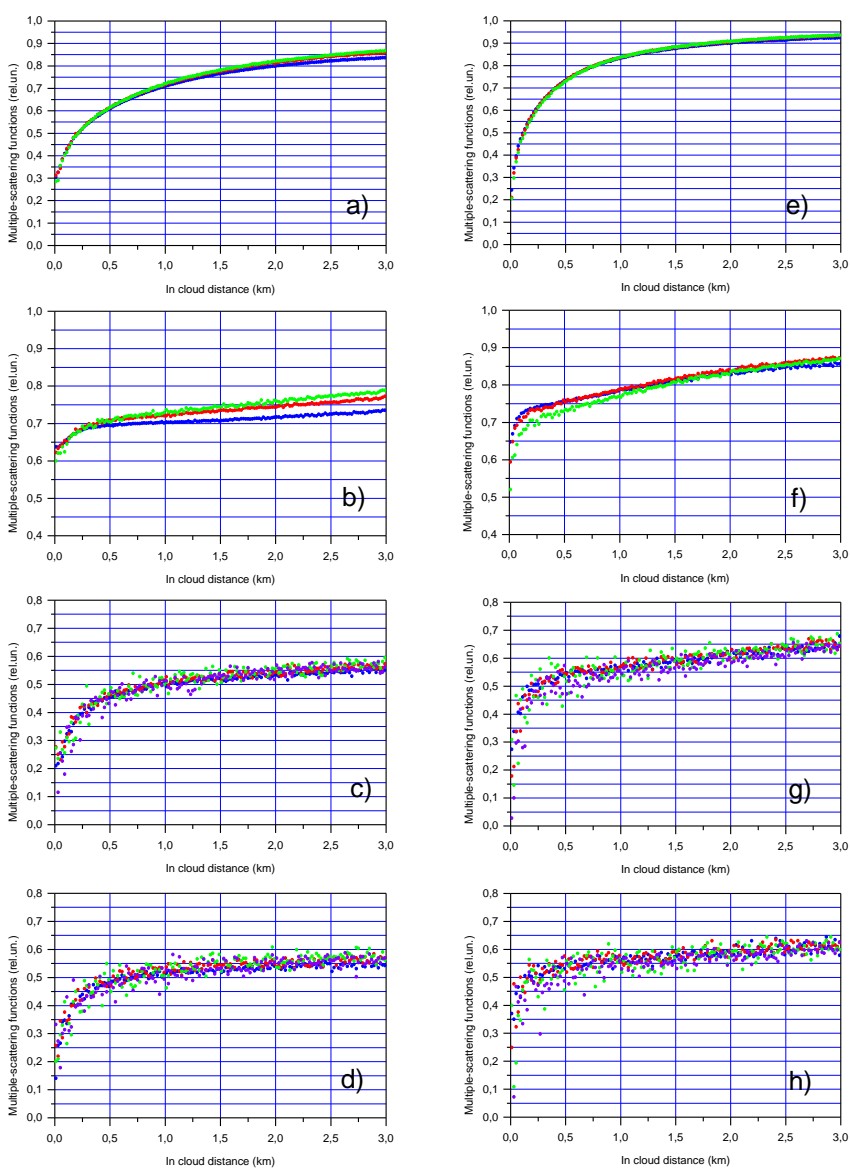

**Figure 8.** MC simulations of multiple scattering functions $\eta_{MS}(d)$. The CALIOP configuration (a – d) and the ATLID configuration (e – h). Coarse-aerosol (a, e), water cloud (b, f), JS cirrus (c, g), Ci cirrus (d, h). The extinction coefficient is of 1.0 km$^{-1}$ (blue points), 0.5 km$^{-1}$ (red points), 0.2 km$^{-1}$ (green points), and 0.06 km$^{-1}$ (purple points).





### 6.1.2 High extinction coefficient of water clouds

In the foregoing subsection, we studied the cases when the values of the extinction coefficient were quite small $\leq 1.0$ km$^{-1}$.
Such a limitation does not conform to warm-cloud properties. Therefore, we performed MC simulation with higher values of
$\varepsilon_p$. We kept most of simulation conditions of the previous section, that is, the CALIOP configuration, the water cloud is within
the altitude range of [8 – 11] km. We employed the refined spatial resolution of 5 m, and 50 orders of scattering were taken
into account.

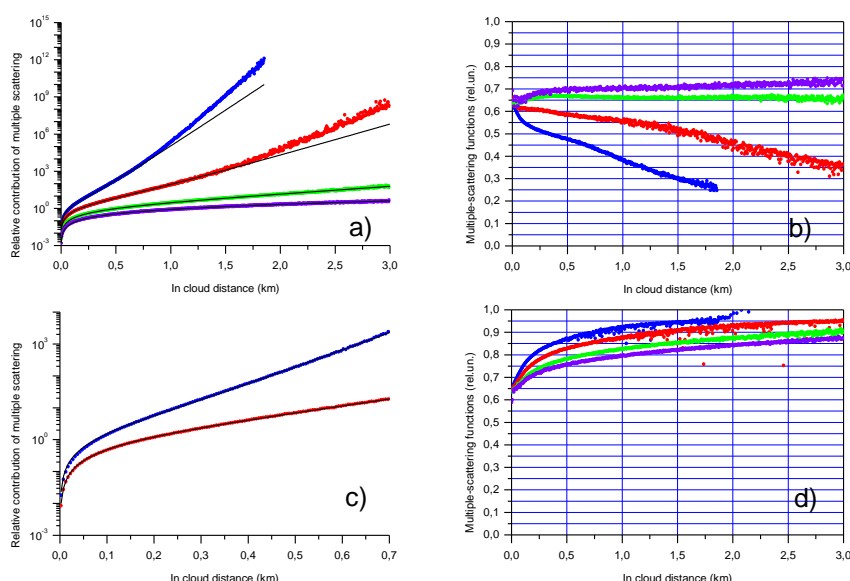


**Figure 9.** Multiple scattering contributions $R_{MSto1}$ to lidar signals (a), points – MC simulations, black lines – fitting with the
empirical model. Panel (c) is same as panel (a) but for the shorter in-cloud range and only two values of the extinction
coefficient are shown. Multiple scattering functions $\eta_{MS}(d)$, multiple scattering (b) and double scattering (d). The extinction
coefficient is of 10.0 km$^{-1}$ (blue points), 5.0 km$^{-1}$ (red points), 2.0 km$^{-1}$ (green points), and 1.0 km$^{-1}$ (purple points).


Figures 9a and 9b show examples of the multiple-scattering effect on lidar signals, i.e., the ratios $R_{MSto1}(d)$ and the MSF
$\eta_{MS}(d)$, respectively. In order to evidence the quality of the fitting, we provide $R_{MSto1}(d)$ within the range $d \in [0; 0.7]$ km in
Fig. 9c. The MSFs $\eta_2(d)$ of the double scattering is shown in Fig. 9d. It was computed according to Eq. 7 while only two
orders of scattering were taken into account. As previously, we maintain the same type of notations in all figures. Blue, red,
green, and purple points show the MC simulation results obtained with the extinction coefficient of 10.0 km$^{-1}$, 5.0 km$^{-1}$, 2.0
km$^{-1}$, and 1.0 km$^{-1}$, respectively. The case $\varepsilon_p = 10.0$ km$^{-1}$ is shown up to the penetration depth of 1.84 km because the





statistical significance of the MC data became extremely low beyond that distance. The black curves represent the fitting results; each curve corresponds to its own set of points $R_{MSto1}(d)$. According to the conclusion of Section 3, only the data that correspond to the penetration optical depth ≤7.0 were taken into account for fitting. It is seen in Figs. 9a and 9c that there is

good agreement between the MC data and the empirical model Eq. (10) when that condition is satisfied. The blue and red points begin to deviate from the corresponding fitting curves at $\tau_p(d) = 7.0$. It should be underlined that the same property is observed for two profoundly different configurations, i.e., CALIOP Fig. 9a and MUSCLE Fig. 2a.

It is instructive to observe the behaviour of the MSF $\eta_{MS}(d)$ in Figs. 8b and 9b. The MSF slightly increases with the in-cloud distance when $\varepsilon_p < 2.0$ km⁻¹, it is almost constant at $\varepsilon_p = 2.0$ km⁻¹, and decreases when $\varepsilon_p > 2.0$ km⁻¹. In other words, the

impact of multiple scattering has the growth rate lower, equal and higher, respectively, than the exponential. All that leads to the large variation of the MSFs values in Fig. 9b. As for the double-scattering MSF $\eta_2(d)$ in Fig. 9d, it always increases. The higher the extinction coefficient $\varepsilon_p$, the higher $\eta_2(d)$ is; to put it differently, the lower the part of the double-scattering is. We note in passing that Figs. 9b and 9d lead to the conclusion that the MSF $\eta_{MS}(d)$ will be overestimated if the number of scattering orders taken into account in MC simulations is deficient. To conclude this subsection, we underline that our empirical

model has successfully fitted the MC data despite the profound changes in the MS growth rate. (The corresponding values of the parameters $\boldsymbol{a} = \{a_1, a_2, a_3\}$ are given in Table S7 of the supplementary material.)

## 6.2 Wide field of view

The multiple-field-of-view technics already has more than four decades history in lidar measurements (see, e.g., Allen and

Platt, 1977; Bissonnette et al., 2005; Jimenez et al., 2020). Multiple scattering impact is favoured by RFOV increasing. Therefore, it is interesting to evaluate the performance of the empirical model against MC simulations when the RFOV is quite wide. The simulations were performed under same conditions as in Subsection 5.1, that is, for the ground-based lidars, the water cloud is within the altitude range of [8 − 11] km, the extinction coefficient is of 1.0 km⁻¹, and 40 orders of scattering were taken into account.

Figures 10a and 10b show examples of the multiple-scattering effect on lidar signals, i.e., the ratios $R_{MSto1}(d)$ and the MSF $\eta_{MS}(d)$, respectively. As previously, we maintain the same type of notations in all figures. Blue, red, green, and purple points show the MC simulation results obtained with the RFOV of 50 mrad, 20 mrad, 10 mrad, and 5 mrad, respectively. (The cases of the RFOV of 0.25 mrad, and 1 mrad are shown in Figs. 3f,b and 4f,b.) The black curves represent the fitting results; each curve corresponds to its own set of points $R_{MSto1}(d)$. It is seen in Fig. 10a that there is good agreement between the MC data

and the empirical model Eq. 10. (The corresponding values of the parameters $\boldsymbol{a} = \{a_1, a_2, a_3\}$ are given in Table S8 of the supplementary material.) As in the cases of high extinction coefficient (see Fig. 9b), there are profound changes in behaviour of the MSF $\eta_{MS}(d)$ when the RFOV increases. All MSFs $\eta_{MS}(d)$ increase with the in-cloud distance at the near end. The MSFs continue increasing for the RFOV of 5 mrad and 10 mrad, i.e., the impact of multiple scattering has the growth rate





lower than the exponential. To the contrary, the MSFs start decreasing at the in-cloud distance about 1 km for the RFOV of 20
mrad and 50 mrad, i.e., the impact of multiple scattering has the growth rate higher than the exponential. All that leads to the
large variation of the MSFs values in Fig. 10b.

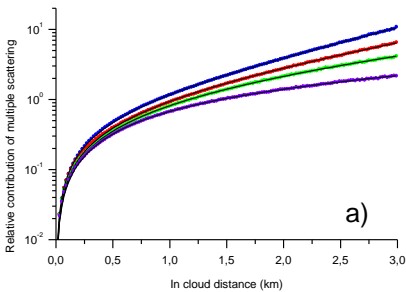
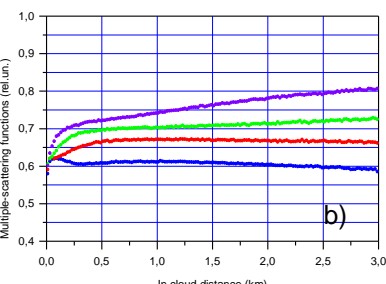

**Figure 10.** Multiple scattering contributions $R_{MSto1}$ to lidar signals (a), points – MC simulations, black lines – fitting with the
empirical model. Multiple scattering functions $\eta_{MS}(d)$ (b). The RFOV is of 50 mrad (blue points), 20 mrad (red points), 10
mrad (green points), and 5 mrad (purple points). The distance to the cloud base is of 8 km.

## 7 Conclusions and discussion

We performed extensive Monte Carlo simulations of single-wavelength lidar signals from a plane-parallel homogeneous layer
of atmospheric particles. The simulations have taken into consideration four types of configurations (the ground based, the
airborne, the CALIOP, and the ATLID) and four types of particles (coarse aerosol, water cloud, jet-stream cirrus and cirrus),
which have large difference in microphysical and optical properties. Most of simulations were performed with the spatial
resolution of 20 m and the particles extinction coefficient between 0.06 km⁻¹ and 1.0 km⁻¹. The resolution was of 5 m for high
values of $\varepsilon_p$ (up to 10.0 km⁻¹). The majority of simulations for ground-based and airborne lidars were performed at two values
of the receiver field-of-view: 0.25 mrad and 1.0 mrad. The effect of the width of the RFOV was studied for the values up to
50 mrad. In order to assure good statistical quality of our Monte-Carlo modelling, each signal was simulated with $4 \cdot 10^{10}$
photons emitted by the lidar (with $4 \cdot 10^{11}$ photons for the cirrus clouds having $\varepsilon_p = 0.06$ km⁻¹).

Such large set of the configurations and the particles characteristics covers a broad range of the multiple-scattering (MS)
relative-contribution to lidar signals: from lower than 5% to several thousands of times. Despite the broad range of variations,
all MS relative-contribution have the same general behaviour as a function of the in-cloud penetration depth when plotted as
a log–linear graph. At the near end, the function $R_{MSto1}(d)$ is linearly proportional to the in-cloud distance $d$. Then, the curve





bends to the right and remains increasing with a same rate within a quite large interval. Such common behaviour enabled us to propose the empirical model, which has demonstrated very good quality of MC data fitting for all considered cases. We

have not confronted any exception despite profound changes in the MS growth rate at high values of the extinction coefficient or wide RFOVs. When R-squared is used to estimate goodness-of-fit (see, e.g., Motulsky and Christopoulos, 2004) to the MC data, all fittings in Figs. (7, 9 – 10) as well as the overwhelming majority in Figs. (3, 5) have $R^2 > 0.99$. Lower values of $R^2$ were obtained for the cases of cirrus clouds in the usual operational conditions.

The fact that the MS relative-contribution can be fitted by a simple function for a large set of lidar configurations and particles

characteristics is of importance by itself. It provides a new perspective on the problem of the radiative transfer related to lidar and radar measurements.

Special attention was given to the usual operational conditions, i.e., when the distance from a lidar to a layer of particles is lower than 15 km, the RFOV $\leq$ 1 mrad, the emitter field-of-view (EFOV) $\leq$ 0.2 mrad, EFOV $\ll$ RFOV, the extinction coefficient $\varepsilon \leq 1$ km$^{-1}$. We assumed the 5% threshold for the MS impact to consider the single scattering approximation as

acceptable. It follows from our Monte Carlo simulations that the multiple scattering effects cannot be neglected when the distance to a particles layer is about 8 km or higher and the full RFOV is of 1.0 mrad. As for the full RFOV of 0.25 mrad, the single scattering approximation is acceptable for aerosols ($\varepsilon_p \lesssim 1.0$ km$^{-1}$), water clouds ($\varepsilon_p \lesssim 0.5$ km$^{-1}$), and cirrus clouds ($\varepsilon_p \leq 0.1$ km$^{-1}$). When the distance to a particles layer is of 1 km, the single scattering approximation is acceptable for aerosols and water clouds ($\varepsilon_p \lesssim 1.0$ km$^{-1}$, both RFOV = 0.25 mrad and RFOV = 1 mrad). As for cirrus clouds, the effect of multiple

scattering cannot be neglected even at such low distance when $\varepsilon_p \gtrsim 0.5$ km$^{-1}$.

As for spaceborne lidars, the contribution of multiple scattering below 5% is so exceptional that the single scattering approximation should be never applied to data of such lidars.

Our simulations confirm general properties of multiple scattering effect on lidar signals that can be found in the literature. Namely, the MS impact as well as the relative contribution of the third and higher orders of scattering increase with increased

extinction coefficient, in-cloud distance, and receiver field-of-view. The proportion of light scattered within forward angles is of upmost importance. Our results suggest that the angle $\theta_{max}$ is more appropriate to characterize that proportion, i.e., for use as one of the parameters that govern the effect of multiple scattering on lidar signals.

We computed the multiple-scattering function $\eta_{MS}(d)$ on the base of our MC data. If follows that $\eta_{MS}(d)$ is a nonlinear function. The assumption $\eta_{MS}(d) = $ const is a rough approximation. It is equivalent to the assumption that the impact of

multiple scattering has an exponential growth rate as a function of the in-cloud optical depth. Generally, this is not the case, especially at the cloud near-end. Moreover, the growth rate as well as the MSF $\eta_{MS}(d)$ depend on the particle extinction coefficient and/or the RFOV all other parameters being the same. In our opinion, the only justification of the assumption $\eta_{MS}(d) = $ const is that it is easily adapted to a solution of an inverse problem.

Despite the fact that this work is limited to the cases of homogeneous layers, we can propose two immediate application of our

results. The empirical model along with the parameters $\boldsymbol{a} = \{a_1, a_2, a_3\}$ given in tables of the supplementary material provide





a fast and accurate way to simulate lidar signals in multiple scattering conditions for a large range of experimental situations. Thus, an interested reader can obtain a set of accurate data without performing time-consuming Monte Carlo simulations.

The first application is that the set of data is used to compute profiles of apparent backscatter, which are employed to test inverse-problem algorithms. Therefore, a developer of an inverse algorithm can see its quality.

The second application is the following. As it was stated in Introduction, the accuracy level and the applicability bounds of the approximate models still need to be rigorously evaluated. Such an evaluation should done in terms of the MS relative-contribution, not in terms of apparent backscatter because a model is devoted to simulate namely the MS effect. The evaluation should be done for a large range of experimental situations. Thus, the results of our work provide an easy way to begin the evaluation.

This work should be considered as the starting stage of the model developing if needs of a practitioner are taken into account, especially when an inverse problem is to be solved (see, e.g., Voudouri et al., 2020). The two next stages have to be fulfilled: (i) to develop an approach that predict $\boldsymbol{a} = \{a_1, a_2, a_3\}$ values only on the base the lidar configuration and particles characteristics; (ii) to generalize $V(d, \boldsymbol{a})$ function to the case of varying profiles of the extinction coefficient.

It seems that the function $V(d, \boldsymbol{a})$ is able to capture the fundamental properties of radiative transfer in the conditions of lidar 605 or radar sounding. Our preliminary results (not shown here) suggest that the empirical model can be fruitful while multiple scattering effects are taken into account in measurements with radars, Raman and high-spectral-resolution lidars as well as in profiles of linear and circular depolarization ratio. Detailed study of empirical-model capacity in those cases is a subject of our future work.

**Appendix A. Multiple-scattering functions**

**A.1 Definitions and relationships**

The utility of a multiple-scattering function (MSF) consists in the possibility to consider effects of multiple scattering while dealing with equations similar to the single scattering lidar equation (1). In what follows, the MSFs are written as functions only of the distance $h$. It should be keep in mind that they depend on particle characteristics and lidar parameters. The notations 615 of Section 3 are used in this Appendix and some relationships of Section 3 are repeated for convenience.

Several approaches to define a MSF can be found in the literature. Similarly to the transport approximation of the radiative transfer theory (see, e.g., Ch.17 of Davison, (1958)), the constant factor $\eta$ was proposed by Platt (1973) to account for "secondary scattering or higher order processes". In the work by Platt (1979), $\eta$ (in our notations the MSF $\eta_{MS}(h)$) was defined as the factor that multiplies the optical depth, has a value less than unity and it may vary with altitude. According to the 620 notations used in the work by Winker (2003), lidar signal that have been corrected for the offset and instrumental factors can be written as:





$$S_{MS}(h) = \left[\beta_p(h) + \beta_m(h)\right] \cdot T_m^2(h) \cdot \exp\left[-2\eta_{MS}(h)\tau_p(h_b, h)\right]. \tag{A1}$$

where $\tau_p(h_b, h) = \int_{h_b}^h \varepsilon_p(h')dh'$ is the in-cloud optical depth, $\varepsilon_p(h)$ is the extinction coefficient of particles.

Another multiple-scattering function $F_{MS}(h)$ was proposed in the work by Kunkel and Weinman (1976) as "a factor, which corrects the extinction coefficient". According to the work by Wandinger (1998) where $F_{MS}(h)$ was employed with explicit separation of molecular and particles characteristics, lidar signal can be written in the form:

$$S_{MS}(h) = \left[\beta_p(h) + \beta_m(h)\right] \cdot T_m^2(h) \cdot \exp\left\{-2\int_{h_b}^h [1 - F_{MS}(h')]\varepsilon_p(h')dh'\right\}. \tag{A2}$$

It is reasonable that the MSFs appeared in Eqs. (A1) – (A2) in the terms related to the particles extinction. The phase function of particles has a sharp forward peak (see examples in Fig. 1). The forward-scattered light remains within the RFOV. Thus, the corresponding optical depth is lower than it has to be in the case of the single scattering. To the contrary, the molecular phase function is smooth. Therefore, a negligibly small fraction of scattered photons remain within the RFOV. We performed MC simulations with the aim to estimate the contribution of multiple scattering to lidar signals from the standard molecular atmosphere. We considered the same lidar parameters as in Section 5.1, i.e., the ground-based lidar with the full RFOV of 1.0 mrad. It follows from our results that the mean value $\langle R_{MSto1}(h)\rangle$ of the molecular contribution was about $4 \cdot 10^{-5}$ for the layer within the altitude range of $[8 - 11]$ km.

Another way to consider the multiple scattering is used in the automated algorithm of the Atmospheric Radiation Measurement Program's Raman lidar (Thorsen and Fu, 2015), i.e., the MSF (in our notations $G_{MS}(h)$) is a factor which corrects the lidar signal of the single scattering approximation:

$$S_{MS}(h) = G_{MS}(h) \cdot \left[\beta_p(h) + \beta_m(h)\right] \cdot T_m^2(h) \cdot T_p^2(h). \tag{A3}$$

As a matter of fact, Eq. (A3) is no more than a mathematical expression that provides an easy way to assign the relationship between $S_{MS}(h)$ and $S_1(h)$. A specific model of multiple scattering appears only when $G_{MS}(h)$ is given in an explicit form. It is obvious that

$$G_{MS}(h) = \frac{S_{MS}(h)}{S_1(h)} = 1 + R_{MSto1}(h) \tag{A4}$$

and properties of $G_{MS}(h)$ are seen directly from our result of the numerical simulations in Sections 5 and 6.

It is a straightforward matter to transform Eqs (A1) and (A2) into the following forms:

$$S_{MS}(h) = \left[\beta_p(h) + \beta_m(h)\right] \cdot T_m^2(h) \cdot T_p^2(h) \cdot \exp\{2[1 - \eta_{MS}(h)]\tau_p(h_b, h)\}, \tag{A5}$$

$$S_{MS}(h) = \left[\beta_p(h) + \beta_m(h)\right] \cdot T_m^2(h) \cdot T_p^2(h) \cdot \exp\left\{2\int_{h_b}^h F_{MS}(h') \cdot \varepsilon_p(h')dh'\right\}, \tag{A6}$$

which lead to the relationships between the Multiple-scattering functions. We have chosen $G_{MS}(h)$ as the known function because MC simulations provide it almost directly.

$$\eta_{MS}(h) = 1 - \frac{1}{2 \cdot \tau_p(h_b, h)} \cdot \ln[G_{MS}(h)], \tag{A7}$$

$$0.5 \cdot \ln[G_{MS}(h)] = \int_{h_b}^h F_{MS}(h') \cdot \varepsilon_p(h')dh', \tag{A8}$$





Equation (A7) repeats Eq. (8). As for Eq. (A8), it is a first-kind Volterra integral equation. The straight way to deduce $F_{MS}(h)$ is numerical differentiation. The problem of numerical differentiation is known to be ill-posed in the sense that small perturbations in the function to be differentiated may lead to large errors in the computed derivative.

An example of the multiple-scattering functions is shown in Fig. A1. The MSFs are drown for the case shown in Figs. 5c and 6c, i.e., the homogeneous cirrus cloud is within the altitude $h$ range of $[8 - 11]$ km, the extinction coefficient is of 1.0 km$^{-1}$, an airborne lidar is at the altitude $h = 7$ km (the distance to the layer base is of 1 km), and the full RFOV is of 1.0 mrad. Random noise is seen in $G_{MS}(h)$, which is inherent in MC simulations. The noise level increases with the penetration into the cloud. The random noise in $\eta_{MS}(h)$ remains acceptable when Eq. (A7) is used. As for $F_{MS}(h)$, errors in the computed derivative were

excessive even when the smoothing with a quite large sliding window was applied. Therefore, the shown in Fig. A1b function $F_{MS}(h)$ was computed using the synergy of the range-dependent smoothing with cubic splines and the method of regularization (see details in Shcherbakov, 2007).

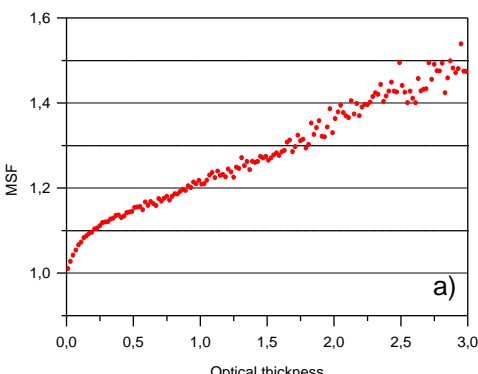

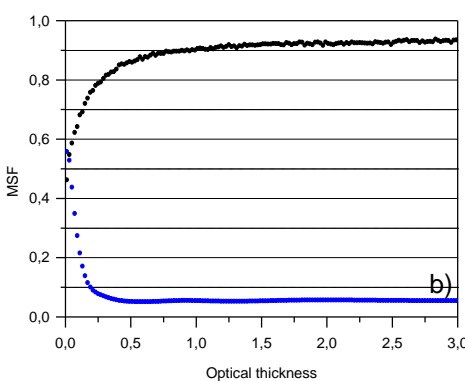


**Figure A1.** Multiple-scattering functions. a) red points $G_{MS}(h)$; b) black points $\eta_{MS}(h)$ and blue points $F_{MS}(h)$.

Some misleading statements about properties of the MSFs can be found in the literature. Thus, to complete this section, we note the following. The functions $\eta_{MS}(h)$ and $F_{MS}(h)$ cannot be approximated by constant values within a quite large range of

the cloud penetration. The function $[1 - F_{MS}(h)]$ should not be confused with $\eta_{MS}(h)$. Their physical meaning is different; the former corrects the extinction coefficient $\varepsilon_p(h)$, whereas the latter affects the optical depth $\tau_p(h_b, h)$. Consequently, the mathematical properties of those functions are different as well.



**A.2 Features at the cloud near-end**


Figures 3 and 5 suggest that the multiple-scattering contribution $R_{MSto1}(h)$ to lidar signals is linearly proportional to $\tau_p(h_b, h)$ at the cloud near-end. This is in agreement with the double-scattering approximation of lidar equation (see, e.g., Samokhvalov, 1979), as well as with the Eloranta model (Eloranta, 1998). The range, when the proportionality is valid, is rather short. Nevertheless, some key features of the MSFs can be found on the base of that approximation. It follows from Eq. (A4)

$\quad G_{MS}(h) \approx 1 + b \cdot \tau_p(h_b, h),$ (A9)

where the coefficient $b > 0$ depends on the phase-function properties. It turns out that $b \approx \mathcal{P}_{2,\pi}/\mathcal{P}_\pi$ (see Ch. 4 Eloranta, 1998), if the Eloranta model is used for simulations.

Equations (A7) and (A9) along with the first two terms of the series expansion of the logarithm $\ln(1 + x)$ lead to the formula:

$\eta_{MS}(h) \approx 1 - 0.5b + 0.25b^2 \cdot \tau_p(h_b, h),$ (A10)

which is in agreement with Eq. (18). We used the condition of a homogeneous cloud $\varepsilon_p(h) = $ const to obtain the approximation of $F_{MS}(h)$ from Eqs. (A8) – (A9) by the analytical differentiation:

$F_{MS}(h) \approx \frac{0.5b}{1 + b \cdot \tau_p(h_b, h)},$ (A11)

It is instructive to see the MSF values exactly on the cloud near-edge, that is, when $h \to h_b$ and $\tau_p(h_b, h) \to 0$:

$\lim_{h \to h_b} G_{MS}(h) = 1,$ (A12)

$\quad \lim_{h \to h_b} \eta_{MS}(h) = 1 - 0.5b,$ (A13)

$\lim_{h \to h_b} F_{MS}(h) = 0.5b.$ (A14)

Intuition suggests that a lidar signal must satisfy the single scattering lidar equation (1) exactly on the cloud near-edge, i.e., when $h = h_b$. That condition and Eq. (A4) impose the value $G_{MS}(h_b) = 1$, which is in agreement with Eq. (A12).

When the multiple-scattering effect is ignored through all the range of distances from a lidar, it is enough to assign $\eta_{MS}(h) = $
1 in Eq. (A5) or $F_{MS}(h) = 0$ in Eq. (A6) and the both equations are reduced to Eq (1). A misleading hypothesis $\eta_{MS}(h_b) = 1$, which is somewhat based on that fact, can be found in the literature when the multiple scattering is supposed to be taken into account. The hypothesis that $\eta_{MS}(h_b) = 1$ is mathematically unjustified, so is the hypothesis that $F_{MS}(h_b) = 0$. As a matter of fact, the condition $S_{MS}(h_b) = S_1(h_b)$ does not impose any restriction on values of $\eta_{MS}(h_b)$ or $F_{MS}(h_b)$, it is fulfilled just due to $\tau_p(h_b, h_b) = 0$. In other words, when $h = h_b$, $\tau_p(h_b, h_b) = 0$ and Eqs. (A5) and (A6) give the same value as Eq (1)
regardless of values of $\eta_{MS}(h_b)$ or $F_{MS}(h_b)$. Therefore, Eqs. (A13) and (A14) are not in contradiction with the intuition suggestion. At the same time, the additional requirement, that the multiple-scattering contribution $R_{MSto1}(h)$ to lidar signals is linearly proportional to $\tau_p(h_b, h)$ at the cloud near edge, imposes the restriction on values of $\eta_{MS}(h_b)$ and $F_{MS}(h_b)$ and leads to Eqs. (A13) and (A14).





## Appendix B. Range-gate effect

It is seen from Eq. (7) that values of $\eta_{MS}$ are very sensitive to chosen values of the optical depth $\tau_p(h_b, h)$ at the distances close to the cloud base. We recall that MC simulations require integration over the range gate. Thus, the question arises: should the value of $\tau_p(h_b, h_i)$ be taken at the far end of the *i-th* range gate or somewhere within it? That question can be answered if Eqs. (1) – (2) are thought of as some mathematical relations where the input parameters can be assigned in an easy-to-use form.

We integrated Eqs. (1) – (2) with the step $\Delta h$ of 20 m to obtain profiles $S_{1,i}$ and $S_{MS,i}$ for the distances $h_i = i \cdot \Delta h; \ i = 1, \dots, N$. The step $\Delta h$ corresponds to the range gate of our MC simulations. The molecular extinction and scattering were neglected. The particulate extinction $\varepsilon_p$ and backscatter $\beta_p$ coefficients as well as $\eta_{MS}$ were assigned to be constant within the whole layer. Thus, we studied effects of the exponential functions of Eqs. (1) – (2). The calculations were performed for a large range of $\varepsilon_p$ values $[1.0; 50]$ km$^{-1}$ and $\eta_{MS}$ within the range of $[0.6; 0.99]$. The estimated values $\tilde{\eta}_{MS,i}(k)$ of the MSF were computed as follows.

$$\tilde{\eta}_{MS,i}(k) = 1 - \frac{1}{2 \cdot [\tau(0, h_{i-1}) + k \cdot \varepsilon \cdot \Delta h]} \cdot \ln\left[\frac{S_{MS,i}}{S_{1,i}}\right], \qquad (B1)$$

were the coefficient $k$ takes values within the range $]0.; 1.]$ and $\tau(0, h_0) = 0$. Equation (B1) corresponds to Eq. (7) with the difference that the value of $\tau_p(h_b, h_i)$ can be taken within the *i-th* range gate. The value of $\tau_p(h_b, h_i)$ is taken at the far end of the range gate when $k = 1$.

A typical example of relative errors $\delta_i(k) = 100 \cdot (\tilde{\eta}_{MS,i}(k) - \eta_{MS})/\eta_{MS}$ of the estimations is shown in Fig. B1. It is seen that $\delta_i(k)$ is close to zero for $k = 0.5$. To the contrary, the relative errors are quite high when $k = 1$ and the optical thickness $\tau_p(h_b, h_i)$ computed from the layer base is rather small. At the same time, $\delta_i(k)$ become negligible with increasing optical thickness of the cloud penetration depth. An example of the effect of the coefficient $k$ on the computed values of the MSF $\eta_{MS}(h_i)$ is shown in Fig. B2. The black points correspond to the case of Fig. 4b above, that is, $h_i$ were assigned to the middle of the range gate ($k = 0.5$). The red points are the MSF $\eta_{MS}(h_i)$ computed using the same MC data but with $k = 1.0$, that is, the values of $\tau_p(h_b, h_i)$ were taken at the far end of the range gate. The discrepancies between the curves are obvious. And what is important, the wrong choice of $\tau_p(h_b, h_i)$, that is, of the coefficient $k$ reverses behaviour of the MSF $\eta_{MS}(h_i)$ at the cloud base.




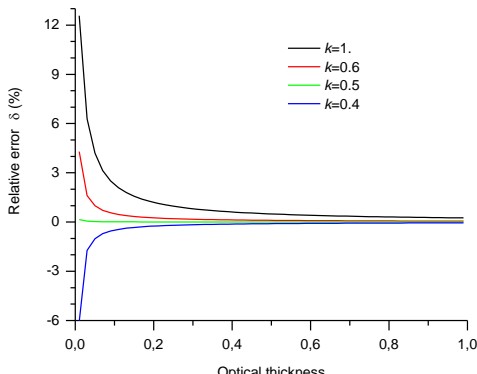

**Figure B1.** Relative errors as functions of the optical thickness from the layer base and the parameter $k$. The green line ($k = 0.5$) and the black line $k = 1.0$) correspond to the cases when $\tau(h)$ is taken in the middle and at the far end, respectively, of the range gate.

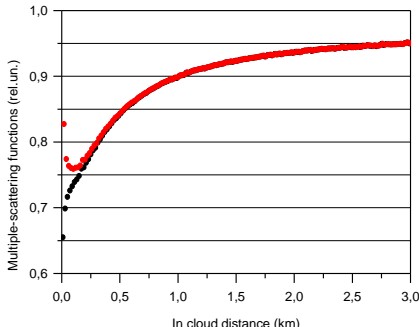

**Figure B2.** Multiple scattering function $\eta_{MS}(h_i)$: $h_i$ assigned to the middle (black points) and to the end (red points) of the range gate.

*Author contributions*

VS developed the empirical model and Appendix A. FS and GM acquired funding. All the authors contributed to developing the McRALI code, numerical simulations, and data treatment. VS wrote the manuscript with help from FS.

*Competing interests*





The authors declare that they have no conflict of interest.

*Acknowledgements*

This work is part of the French scientific community EECLAT project (Expecting EarthCARE, Learning from A-train). The EECLAT community and research activities are supported by the National Center for Space Studies (CNES) and the National Institute for Earth Sciences and Astronomy (INSU).

*Financial support*

This research has been supported by the National Institute for Earth Sciences and Astronomy (INSU grant).

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
