# Peer review of "Empirical model of multiple scattering effect on single-wavelength lidar data of aerosols and clouds"

_Atmospheric Measurement Techniques, 2021_

## Author Comment (AC1)

Response to Reviewer # 3

We thank the reviewer for his review and valuable comments. The manuscript has been modified according to the suggestions proposed by the reviewer. The remainder is devoted to the specific response item-by-item of the reviewer's comments.

We have answered to all reviewer's comments. Some comments, mostly concerning general aspects of the Monte Carlo method or mathematics, are addressed below but did not lead to revisions in the text of the manuscript.

*RC=Reviewer Comments*
AR=Author response
TC=Text Changes

**General Comments**

*The paper presented by Shcherbakov et al. aims to establish an empirical relationship to account for the effects of multiple scattering on lidar measurements from the ground, aircraft and satellite. As stated by the authors, there have already been many studies to take this effect into account. Some have directly used Monte Carlo modelling; others have used radiative transfer codes allowing the development of phase functions over several orders. From these previous studies, parameterizations have already been developed, in particular to take into account the forward scattering which is preponderant in the multiple scattering for large particles.*
*The proposed parameterization is based on cases where the optical thickness (OT) remains below 7, which is indeed an upper limit for the vast majority of lidars. It explores mainly FOVs of 0.25 and 1 mrad which may appear somewhat limiting for existing groundbased and airborne lidar systems. The contribution of this study is not sufficiently demonstrated. It is clear that the definition of a lidar system is generally carried out taking into account the constraints linked to the observation geometry and that Monte Carlo simulations are performed by the designers, as was done for CALIOP and is done for airborne and ground-based lidars. This aspect of Monte Carlo simulation is therefore not original in itself and many models exist in laboratories around the world. It is a basic design tool.*
We agree with the reviewer that several models exist in laboratories around the world, those models take into account the forward scattering, which is preponderant in the multiple scattering for large particles.
The problem is that all fast models without exception are based on some approximations to take into account multiple scattering (MS), and the accuracy level and the applicability bounds of the approximate models are not well addressed in the literature. The later statement is taken from the workshop report "INSTRUMENTS ACTIFS" issued by the participants of the "Transfert Radiatif dans les ATmosphères Terrestres pour les ObseRvations spatIAles" "TRATTORIA 2020" (see the URL: http://www.meteo.fr/cic/meetings/2020/trattoria/, accessed 22.12.2021), (unfortunately the report is not available by the internet).
We performed a huge set of Monte-Carlo (MC) simulations for a large range of experimental situations and reported in this work **quantitative** data of the MS effect. It is demonstrated that the proposed empirical model is in very good agreement with the quantitative data. Therefore, the accuracy of any fast approximate model can be evaluated without performing time-consuming MC simulations, i.e., just using the empirical model along with the parameters $\boldsymbol{a}=\{a1,a2,a3\}$ given in tables of the supplementary material.

For example, our preliminary results (not shown here) suggest that the model by Eloranta (1998) (EM) adequately reproduces the profiles of relative contribution of 2nd, 3rd, and 4th orders of scattering (the case of water cloud, a ground based lidar, the usual operational conditions) **providing** that values of the EM parameters are **adjusted**, especially, the values of $\mathcal{P}_{n,\pi}/\mathcal{P}_{\pi}$ ($n \geq 2$), which are usually lower than 0.5. (We recall that according to the work by Eloranta (1998), "for typical phase functions, $\mathcal{P}_{n,\pi}/\mathcal{P}_{\pi}$ is between 0.5 and 1".)

*There are several points that need to be better explained in this article, such as*

*1) The justification of the optical properties considered as initial conditions for the modelling. Are they common, extreme?*
In response to the pertinent questions of the reviewer, we have added to the revised manuscript Chapter 5.3 (page 21, line 455).
5.3 Estimation of MS magnitude in other cases

[revised manuscript text omitted]

*3) Better justify the choice of an altitude interval of 8 to 11 km which is not optimal for aerosol layers. It would have been preferable to choose realistic layers such as a Saharan boundary layer that can extend from the surface to 3-5 km above.*
Knowing that the distance to a layer is one of the key parameters, which govern the effect of MS on lidar signals, and the contribution of the molecular scattering to MS can be neglected, we deliberately placed all four types of particles at the same distance from a lidar, i.e., within the same altitude range. With such a choice, the phase-function impact is put in the forefront. The results of the Sections 5 and 6 are presented so that they remain unaltered when the lidar pointing angle and/or the layer altitude vary provided that the distance to the cloud base/border remains unchanged. We have added to the revised manuscript (page 4, line 128) the following text.

For example, if a Saharan boundary layer extends from the surface to the range $3 - 4.12$ km above and a ground based lidar is tilted by 68 degrees with respect to the zenith, the curves of Figs. 3(a, e) can be used to assess MS effects.

*4) Clarify the results on* $\eta(h)$*, the parameter initially defined by Platt. At the entrance to the scattering layer, the multiple scattering is small and* $\eta_{MS} = 1$ *(Eq. 7), but this is not the case when looking at the figures. Is it really the in-cloud distance that is on the x-axis? Or maybe I have misunderstood, and better explanations are needed.*

Unfortunately, the statement that $\eta_{MS}(h_b) = 1$ at the entrance to the scattering layer can be found in the literature. That statement is **not correct** as it is demonstrated in Appendix A.2 of our work. In what follows, we demonstrate that result again this time using the model by Eloranta (1998).
We have to underline that the equation

$$\eta_{MS}(h) = 1 - \frac{1}{2\cdot\tau_p(h_b,h)} \cdot \ln\left[\frac{S_{MS}(h)}{S_1(h)}\right] \qquad (7)$$

is an **undefined** algebraic expression (the indeterminate form "0/0") at the entrance $h_b$ to the scattering layer, i.e.,

$$\lim_{h \to h_b} \tau_p(h_b, h) = 0, \ \lim_{h \to h_b} \frac{S_{MS}(h)}{S_1(h)} = 1, \text{ and } \lim_{h \to h_b} \ln\left[\frac{S_{MS}(h)}{S_1(h)}\right] = 0, h \geq h_b.$$

The evaluation of the undefined expression above needs correct treatment in mathematical terms of the statement that the multiple scattering contribution to a lidar signal is small.
Equation (12) of the work by Eloranta (1998) leads directly to the expression:

$$\frac{S_{MS}(h)}{S_1(h)} = 1 + \sum_{n=2}^{\infty} \frac{\mathcal{P}_{n,\pi}}{\mathcal{P}_{\pi}} \cdot \frac{[\tau_p(h_b,h)]^{n-1}}{(n-1)!}$$

Accordingly, we can write using the first term of the expansion in powers of $\ln(1 + x)$, i.e., of $\ln\left[\frac{S_{MS}(h)}{S_1(h)}\right]$

$$\eta_{MS}(h) = 1 - 0.5 \cdot \frac{\mathcal{P}_{2,\pi}}{\mathcal{P}_{\pi}} - \frac{1}{2} \cdot \sum_{n=3}^{\infty} \frac{\mathcal{P}_{n,\pi}}{\mathcal{P}_{\pi}} \cdot \frac{[\tau_p(h_b,h)]^{n-2}}{(n-1)!}$$

It directly follows that

$$\eta_{MS}(h_b) = \lim_{h \to h_b} \eta_{MS}(h) = 1 - 0.5 \cdot \frac{\mathcal{P}_{2,\pi}}{\mathcal{P}_{\pi}}$$

That result is in total agreement with Eq. (A13) of Appendix A.2 of our work.
Generally speaking, $\eta_{MS}(h_b) < 1$ at the cloud near-end. That property is the direct outcome of the fact that the MS is proportional to the optical thickness at small values of $\tau_p(h_b, h)$.
We have added to the revised manuscript (page 15, line 378) the following text.
Generally, the multiple-scattering functions $\eta_{MS}(d)$ in Figs. 4, 6, and 8 reveal the same property at the layer near-end, i.e., $\lim_{d \to 0} \eta_{MS}(d) < 1$. That result is in total agreement with the theory (see Appendix A.2 below).

*5) The results should be presented in a more synthetic way in order to lighten the reading of the article. It could be interesting to make 2D figures* $\eta MS = f (FOV, OT)$*.*
A study of the effect of RVOV **in detail** is not the subject of this work. We performed MC simulation for a few values of RFOV. Therefore, our data do not provide possibility to make such 2D figures.

*6) It is difficult to properly assess the robustness of the parameterization. In the case of CALIOP corrections are applied and it would be interesting to compare them to those proposed via the parametrization.*

Our Monte Carlo results and the empirical model are within the domain of direct problems. We recall that good quality solutions of a direct problem are necessary to test inverse-problem algorithms. Our empirical model provide means to test existing inverse-problem algorithms without MC simulations.

Algorithms of correction and inversion of signals of space born lidars belong to the domain of inverse problems. Inverse problems are not the subject of this study. We are not ready to answer to the question of CALIOP corrections at this very moment. A lot of additional work has to be done.

*Other aspects:*

*P4. How are the number of realizations (photons) chosen? How are the results degraded?*

The sample size of our modelling (the number of realizations (photons)) was limited by our computing capacities. The results were not deliberately degraded.

"Light propagation can be regarded as a Markov chain of photon collisions in a medium in which it is scattered or absorbed. The Monte Carlo technique consists in computational simulation of that chain and in calculating a statistical estimate for the desired functionals" (Ch. 1.1, Marchuk et al., 1980). Our results were not degraded; the statistical error of the estimate (i.e., random noise or the spread of points) are inherent in MC simulations.

The statistical error of the estimate decreases with the **square root** of the sample size, i.e., of the number of realizations (photons). In other words, the higher the number of realizations, the lower the random noise is. In order to assure good statistical quality of our Monte-Carlo modelling, each signal was simulated with very high number of realizations (photons), say, at the limit of our computing capacities.

For example, one case with $4\cdot10^{10}$ photons emitted by the lidar takes about 18 hours of the computing time ("DELL PowerEdge R940 Server" with 20 jobs running in parallel); $4\cdot10^{11}$ emitted photons takes about 180 hours. It would be preferable to reduce the random noise by 5 times in the cases of cirrus particles, but it would take about $180 \cdot 5^2 = 4500$ hours, which is not reasonable.

*P5. Degassing feathers = ash feathers?*

We added to the revised manuscript (page 5, line 159) the following text.

… for ash particles in volcanic degassing plumes …

*Table 1. Define the parameters in the caption table.*

Corrected in the revised manuscript.

*P8. The number of significant scattering orders is closely related to the number of photons. It would be nice to see this relationship.*

There exists a set of parameters that predetermine the number of photons required for MC simulations. The number of significant scattering orders is one of those parameters. In our opinion, a huge set of MC simulations has to be done just to get some idea about such relationship (if it exists). We recall that MC simulations are very time consuming.

*P10L160. For figs 3,5 and 8 it is R that is used while the adjustment is made on G. In order to make it more understandable for the reader, it would be better to present G or to adjust on R. At this stage of the paper, there is nothing to justify such a model. Why this choice?*

The function $G_{MS}(h)$ is chosen as the empirical model because Eq. (2) has the same structure as the lidar equation (1). The relative contribution of multiple scattering $R_{MSto1}(h)$ has the

simple relationship with $G_{MS}(h)$ (see Eq. (9)). It has clear meaning; that is why it is frequently used for graphical representation of data in the literature.

*Eq.18. Development to 2nd order of arctan whereas before it was 1st order*
As usual in mathematics, we write the series expansion till the first non-zero term with respect to the variable. Thus, it is seen and it is of importance that the term of 1st order with respect to the in-cloud distance is absent in Eq. (18).

*P11L292. This is an important result for the paper and should be in the body of the paper or at least in an appendix. Furthermore, it would have been nice to show graphically the fits on an example and a sensitivity study to these fits against the coefficients a1,a2 and a3.*
We hope that the values of the fitting parameters $\boldsymbol{a}$={$a1,a2,a3$}, among other results of this work, will be used by the scientific community.
On the other hand, the specific values of $\boldsymbol{a}$={$a1,a2,a3$} correspond to the particular cases. Tables of values will overload the text, and we do not perceive the specific values among main results of this work. Therefore, we prefer to provide an interested reader with the specific values through the supplementary material.

*Figure 3: What explains the difference in the spread of points between the layer types?*
We added to the revised manuscript (page 12, line 321) the following text.
The difference in the spread of points between the layer types is clearly seen in Figs. (3) – (8). It is in agreement with the general property of MC simulations of radiative transfer (see, e.g., Buras and Mayer, 2011). The stronger forward peak of the scattering phase function, the slower convergence of MC simulations is. In other words, the lower value of $\theta_{max}$, the higher the spread of points is (all other parameters being the same).

*P14L330. Why 5%?*
We added to the revised manuscript (page 14, line 344) the following text.
It follows from EARLINET (European Aerosol Research Lidar Network) instrument intercomparison campaigns (Fig. 4b, Wandinger et al., 2016) that the relative deviation of the lidar signals ($\lambda = 0.532$ µm) from the common reference is mostly within ±3%. In our opinion, MS contribution lower than 5 % could hardly be detected in such conditions. It should be underlined that the results of this work are presented so that an interested reader can use other threshold value to assess whether the single scattering approximation is acceptable in view of measurement errors of a specific lidar.

*P14L336. 8 km of higher = 8 km or higher?*
Corrected in the revised manuscript.
*Table 2. The values in the table are in %?*
Tables (2) – (5) are entitled "Multiple scattering contribution to lidar signals in **percent** …". Thus, the values are in %.
Corrected in the revised manuscript.

*Appendix A. It is not clear what the authors are trying to demonstrate here. The functions are indeed different, but they remain related. It depends on the initial hypothesis that is considered. Is this appendix useful?*
Unfortunately, some misleading statements about properties of the multiple-scattering functions can be found in the literature. For example, (i) the function $[1 - F_{MS}(h)]$ is confused with $\eta_{MS}(h)$; (ii) it is stated that $\eta_{MS} = 1$ at the entrance to the scattering layer. (We do not

provide citations out of ethical considerations. An interested reader can easily find such papers by himself/herself.)
In our opinion, it is of importance to bring together the correct relationships. It could be said that Appendixes of this work are mostly addressed to PhD students.

---

## Author Comment (AC3)

Response to Reviewer # 1

We thank the reviewer for his review and valuable comments. The manuscript has been modified according to the suggestions proposed by the reviewer. The remainder is devoted to the specific response item-by-item of the reviewer's comments.

*RC=Reviewer Comments*
AR=Author response
TC=Text Changes

*This paper develops simple functions to characterize the impacts of multiple scattering on lidar observations, based on simulations from a physics-based Monte Carlo multiple scattering code. The simulations are performed for one type of coarse aerosol, one water cloud case, and two cirrus cases, for typical configurations of ground-based and airborne lidars and for the CALIOP and ATLID spaceborne lidars.*

*I think this paper is a useful introduction to and overview of lidar multiple scattering effects. I disagree with the comment from RC2, who says "This aspect of Monte Carlo simulation is therefore not original in itself and many models exist in laboratories around the world. It is a basic design tool." Not every lidar group considers multiple scattering or applies corrections. Multiple scattering codes should be a basic design tool, but it is often not considered in lidar retrievals under an assumption that the lidar design ensures they are insignificant. The results presented in this paper are helpful to groups which haven't previously considered multiple scattering, and to users of lidar data who want to understand under what conditions the impacts should be considered and perhaps apply corrections to the data which is not already corrected.* We are grateful to the reviewer for providing his opinion, which we fully share. Among other things, the above arguments motivated this work.

*The discussion of the method is sufficiently detailed but the discussion of the results is mostly a factual description of the simulations and fitting results. Some interpretation and synthesis of the results into general conclusions and guidance is needed. The major goal of the paper seems to be to identify conditions where multiple scattering are small enough they can be ignored. The paper identifies these conditions for the four particle types considered and two 'standard' FOVs. The authors should use their results to make more general statements. Only a few specific lidar viewing geometries and particle cases are considered.* In response to the pertinent questions of the reviewer, we have added to the revised manuscript Chapter 5.3 (page 21, line 455) (see below).

*What are the limitations in using these fitting equations to estimate multiple scattering for other conditions (range, FOV, extinction, particle size).* We already underlined in the section "Conclusions and discussion" that the empirical model has demonstrated very good quality of MC-data fitting for all considered cases. We have not confronted any exception despite profound changes in the MS growth rate at high values of the extinction coefficient or wide RFOVs.
It seems that our empirical model has no limitations from point of view of the fitting quality of MS contribution to lidar signals provided that MC simulations were performed and the values of the coefficients $\boldsymbol{a}=\{a1,a2,a3\}$ were found. We also underlined in the section "Conclusions and discussion" that an approach has to be developed to predict $\boldsymbol{a}=\{a1,a2,a3\}$ values only on the base the lidar configuration and particles characteristics, and that the empirical model has to be generalized to the case of varying profiles of the extinction coefficient.

*Are the aerosol, water cloud, and cirrus types defined in a way that they predict typical multiple scattering effects? Are the conclusions valid over expected variations in particle size? There is some variability in cirrus phase functions due to differences in particle habit. Would you expect variations in habit to change these conclusions?*

In response to the pertinent questions of the reviewer, we have added to the revised manuscript Chapter 5.3 (page 21, line 455).

5.3 Estimation of MS magnitude in other cases

[revised manuscript text omitted]

**Specific comments**

*Line 23: The authors should quantify here what is meant by "acceptable"*
We added to the revised manuscript (page 1 line 23) the following text.
…, i.e., multiple scattering contribution to lidar signal is lower than 5% …

*Line 126: Is "coarse aerosol" meant to represent dust? More details should be provided on the model for coarse aerosol: index of refraction, shape (spheres, spheroids, aspect ratio), and size. Why was this particular model chosen, is it generally representative of coarse aerosol? Is multiple scattering different for desert dust or hydrated sea salt aerosol of similar size? How sensitive are the results to changes in aerosol optical properties?*
We added to the revised manuscript (page 5 line 146) the following text.
The scattering matrix of the coarse-aerosol was simulated according to the work by Dubovik et al. (2006) as the "Mixture 1" of spheroids with the distribution of axis ratios within the range [0.3349, 2.986] (assuming, as the first-order approximation, that shape is independent of size). The size distribution of particles was assumed to be log-normal with the mean radius of 2 μm, the standard deviation of 0.6 μm, and $d_{eff} = 4.75$ μm. That value is in agreement with data of the work by Weinzierl et al., (2009), where it was found that the effective diameter of the Saharan dust showed two main ranges: around 5 μm and 8 μm. The real and imaginary part of the refractive index were 1.55 and 0.002, respectively (see, e.g., Petzold et al., 2009).

*Line 321: Explain why 5% is selected as the threshold where the multiple scattering contribution must be considered. Because 5% is smaller than other sources of error typically found in lidar retrievals?*
The threshold 5% was chosen from point of view of measurement errors. We agree that 5% is smaller than other sources of error that affect lidar retrievals. We added to the revised manuscript (page 14 line 344) the following text.
It follows from EARLINET (European Aerosol Research Lidar Network) instrument intercomparison campaigns (Fig. 4b, Wandinger et al., 2016) that the relative deviation of the lidar signals ($\lambda = 0.532$ μm) from the common reference is mostly within ±3%. In our

opinion, MS contribution lower than 5 % could hardly be detected in such conditions. It should be underlined that the results of this work are presented so that an interested reader can use other threshold value to assess whether the single scattering approximation is acceptable in view of measurement errors of a specific lidar.

*Technical corrections*
We are grateful to the reviewer for providing the technical corrections.

*Line 24 and 40: "of 1 km" does this mean 'equals 1 km', 'less than 1 km'?*
Corrected in the revised manuscript.
*Line 65: "techniques" should be "technique"?*
Corrected in the revised manuscript.
*Line 77: when "the" impact*
Corrected in the revised manuscript.
*Line 100: "The" other two ...*
Corrected in the revised manuscript.
*Line 265, 383, 409, and 452: "again" rather than "another time"*
Corrected in the revised manuscript.
*Line 632: "drown" should be "shown"?*
Corrected in the revised manuscript.
*Line 637: "the shown in Fig. A1b function" should be "the function shown in Fig. A1b", I think*
Corrected in the revised manuscript.